# Structure of the AlgKX modification and secretion complex required for alginate production and biofilm attachment in *Pseudomonas aeruginosa*

Andreea A. Gheorghita [1,2], Yancheng E. Li [1,2,5], Elena N. Kitova [3], Duong T. Bui [3], Roland Pfoh [1], Kristin E. Low [1,6], Gregory B. Whitfield [1,2,7], Marthe T. C. Walvoort [4,8], Qingju Zhang [4,9], Jeroen D. C. Codée [4], John S. Klassen [3] & P. Lynne Howell [1,2] ✉

Synthase-dependent secretion systems are a conserved mechanism for producing exopolysaccharides in Gram-negative bacteria. Although widely studied, it is not well understood how these systems are organized to coordinate polymer biosynthesis, modification, and export across both membranes and the peptidoglycan. To investigate how synthase-dependent secretion systems produce polymer at a molecular level, we determined the crystal structure of the AlgK-AlgX (AlgKX) complex involved in *Pseudomonas aeruginosa* alginate exopolysaccharide acetylation and export. We demonstrate that AlgKX directly binds alginate oligosaccharides and that formation of the complex is vital for polymer production and biofilm attachment. Finally, we propose a structural model for the AlgEKX outer membrane modification and secretion complex. Together, our study provides insight into how alginate biosynthesis proteins coordinate production of a key exopolysaccharide involved in establishing persistent *Pseudomonas* lung infections.

*Pseudomonas aeruginosa* is an opportunistic human pathogen that is known to cause chronic lung infections in individuals with cystic fibrosis (CF), and, more recently, superinfections in COVID-19 patients[1,2]. The establishment of *P. aeruginosa* infection in the CF lung is a vital turning point in disease severity and the leading cause of patient morbidity and mortality[3–6]. In chronic CF infections, *P. aeruginosa* overproduces the alginate exopolysaccharide, a notorious virulence factor that aids in the evasion of the host immune

response and increases tolerance to antibiotics[7,8]. During COVID-19 pneumonia, *P. aeruginosa* undergoes a rapid adaptive evolution activating alginate production[1]. The adapted *P. aeruginosa* display enhanced persistence in the lung and promotes superinfections that influence disease severity in COVID-19 patients[1,9].

Alginate is produced by a synthase-dependent exopolysaccharide secretion system—a conserved molecular mechanism in Gram-negative bacteria for manufacturing and exporting carbohydrate

[1]Program in Molecular Medicine, The Hospital for Sick Children, Toronto, ON, Canada. [2]Department of Biochemistry, University of Toronto, Toronto, ON, Canada. [3]Department of Chemistry, University of Alberta, Edmonton, AB, Canada. [4]Leiden Institute of Chemistry, Leiden University, Leiden, The Netherlands. [5]Present address: Division of Chemistry and Chemical Engineering, California Institute of Technology, Pasadena, CA, USA. [6]Present address: Lethbridge Research and Development Centre, Agriculture and Agri-Food Canada, Lethbridge, AB, Canada. [7]Present address: Département de Microbiologie, Infectiologie et Immunologie, Université de Montréal, Montréal, QC, Canada. [8]Present address: Department of Chemical Biology, Stratingh Institute for Chemistry, University of Groningen, Groningen, The Netherlands. [9]Present address: National Research Centre for Carbohydrate Synthesis, Jiangxi Normal University, Nanchang, China. ✉e-mail: howell@sickkids.ca

**Fig. 1 | Structure of the AlgKX complex. a** Schematic of the alginate exopolysaccharide. Alginate is a random polymer composed of 1–4 linked α-ʟ-guluronate (GulA) residues and ß-ᴅ-mannuronate (ManA) residues. ManA residues can be *O*-acetylated (Acetylated ManA). **b** Depiction of proteins involved in the alginate biosynthesis synthase-dependent secretion system organized by function. The PilZ domain of Alg44 (PDB: 4RT0)[16] binds to c-di-GMP in the cytoplasm, triggering ManA polymerization by Alg8 from GDP-ManA and its translocation across the inner membrane (IM). Once in the periplasmic space, AlgI, AlgJ (PDB: 4O8V)[19], AlgF (PDB: 6CZT), and AlgX (PDB: 4KNC)[18] modify ManA residues by *O*-acetylation. AlgG (PDB: 4OZZ)[20] epimerizes unacetylated ManA residues to GulA[13]. AlgL (PDB: 4OZV) degrades alginate that accumulates in the periplasmic space[25]. AlgK (PDB: 3E4B)[23]

and AlgE (PDB: 3RBH)[39] are involved in the export of alginate from across the outer membrane (OM). **c** Complex of *P. putida* AlgK (light green) and AlgX (periwinkle). **d** Close-up of the AlgKX_*Pp* interaction interface with hydrogen bonds and salt bridge interactions represented by yellow and pink lines, respectively. The N and C-termini are represented by N and C, respectively and colored as defined in **c**. **e** Conservation of residues involved in the AlgKX_*Pp* interaction as calculated by ConSurf[65]; green indicates residues that are variable and less conserved, and purple indicates residues that are highly conserved. Residues that are underlined represent main chain interactions and residues that are italicized represent side chain interactions. Chloride (large yellow-filled circle), nickel (small green-filled circle), and glycerol (beige, stick representation) are observed in panels **c**, **d**, and **e**.

polymers, including cellulose, acetylated cellulose, poly-ß-ᴅ-*N*-acetylglucosamine (PNAG), Pel, and alginate[10–12]. Synthase-dependent systems are characterized by a membrane-embedded glycosyl transferase involved in polymer synthesis, a cyclic-di-GMP binding protein that regulates polymer synthesis, one or more periplasmic polymer-modifying enzymes, an outer membrane-linked protein with a tetratricopeptide repeat (TPR)-containing domain, and a ß-barrel porin through which the polymer is exported[10–12]. In the *P. aeruginosa* alginate biosynthetic system, alginate is synthesized by the glycosyl

transferase Alg8 and cyclic-di-GMP receptor Alg44 as a ᴅ-mannuronate (ManA) homopolymer before it is translocated to the periplasm (Fig. 1a, b)[13–17]. After translocation, the polymer is modified either by acetylation by the concerted action of AlgI, AlgJ, AlgF, and the terminal acetyltransferase AlgX[18,19], or by epimerization to ʟ-guluronate (GulA) by AlgG[13,20] (Fig. 1a, b). The degree of alginate acetylation can vary from 4 to 57%, depending on the strain of *Pseudomonas*, its growth conditions, and the amount of ManA within the polymer[21,22], while the degree of alginate epimerization is not well-characterized. To date, the order

in which these modifications occur is still unknown. After polymer modification, mature alginate is exported into the extracellular milieu via the TPR-containing outer membrane lipoprotein AlgK and outer membrane β-barrel porin AlgE[23,24]. The alginate lyase AlgL maintains the homeostasis of the periplasm by degrading accumulated alginate[25] (Fig. 1b).

Although interactions between proteins within synthase-dependent secretion systems have been identified, there is a gap in our understanding of how the processes of polymerization, modification, and export are coordinated for exopolysaccharide production. In this study, we focus on the alginate proteins AlgX and AlgK involved in polymer acetylation and export, respectively. An interaction between AlgX and AlgK has been previously reported in *P. aeruginosa*[26], however, how these two proteins interact and the consequences of this interaction on alginate production have not been investigated. In this study, we present a co-crystal structure of a polymer-modifying enzyme and a TPR-containing protein in a synthase-dependent exopolysaccharide biosynthesis system. The AlgKX protein complex reveals the molecular details of a crucial protein–protein interaction involved in alginate production and provides fundamental insight into how a polymer modification enzyme and export protein function collaboratively in synthase-dependent secretion systems. Using electrospray ionization mass spectrometry (ESI-MS), we establish that AlgKX directly binds alginate polymers of various lengths and compositions. We demonstrate that the formation of AlgKX is vital for alginate production and biofilm attachment in *P. aeruginosa*. Finally, we propose a model for the AlgEKX outer membrane alginate modification and export complex.

## Results

### Structure of the AlgKX complex
To understand how AlgK and AlgX interact at a molecular level, we co-crystallized *Pseudomonas putida* AlgK without its native signal sequence (AlgK$_{Pp}^{33-484}$) and full-length *P. putida* AlgX (AlgX$_{Pp}^{1-479}$) (Supplementary Figs. 1, 2) and determined the structure of the AlgKX$_{Pp}$ complex to 2.5 Å resolution (Fig. 1c, Supplementary Table 1, and Supplementary Fig. 3). The AlgKX$_{Pp}$ complex crystallized in space group *I* 4 2 2 with a single copy of AlgK$_{Pp}$ and AlgX$_{Pp}$ in the asymmetric unit. In the isolated structures used for molecular replacement, *Pseudomonas fluorescens* AlgK (AlgK$_{Pf}$) contains 9.5 TPR motifs (1/2 of TPR R1 and R2-R10)[23], while AlgX$_{Pa}$ contains an N-terminal SGNH hydrolase-like domain (residues 42–347) and a C-terminal carbohydrate-binding module (residues 348–463)[18]. In the AlgKX$_{Pp}$ complex, we were able to model eight TPR motifs (R3-R10) in AlgK$_{Pp}$ (residues 138–452) and residues 29–468 of AlgX$_{Pp}$ (Fig. 1c). The N-terminus of AlgX$_{Pp}$ (residues 29–37) adopts an extended conformation in the complexed structure that could not be modeled in the isolated AlgX$_{Pa}$ structure[18], suggesting that the extended N-terminus of AlgX may be disordered and dynamic in nature, thus requiring interaction with AlgK for stabilization (Supplementary Fig. 4).

The AlgKX$_{Pp}$ structure reveals that it's mainly the N-terminus of AlgX$_{Pp}$ (residues 30–37) that interacts with TPRs R9-R10 of AlgK$_{Pp}$, however additional residues outside the N-terminus of AlgX$_{Pp}$ also interact with AlgK$_{Pp}$ (Fig. 1d). Using the Proteins, Interface, Surfaces, and Assemblies (PISA) server[24], the solvation energy and total binding energy of this interface were calculated to be −7.874 and −13.57 kcal/mol, respectively, suggesting that the interface we observe is biologically relevant and not an artifact of crystal-packing. PISA also calculated the interaction interface to have a buried surface area of 1015 Å² mediated by 12 hydrogen bonds and one salt bridge. Within the N-terminus of AlgX$_{Pp}$, the main chains of P30, F32, and A34 hydrogen bond to the side chains of AlgK$_{Pp}$ R383 and Q372. The side chains of AlgX$_{Pp}$ E35 and AlgK$_{Pp}$ R404 form a salt bridge, while the main chains of these residues interact via hydrogen bonding. The main chains of AlgX$_{Pp}$ C37 and AlgK$_{Pp}$ G402 also interact via hydrogen bonding.

Outside of the AlgX$_{Pp}$ N-terminus, the side chain of S234 and the main chain of Y235 interact with AlgK$_{Pp}$ H401 (Fig. 1d). The side chain of AlgX$_{Pp}$ R174 interacts with the side chain of AlgK$_{Pp}$ N368 and the main chain of G367 via hydrogen bonding. The side chain of AlgX$_{Pp}$ K213 interacts by hydrogen bonding with the main chain of AlgK$_{Pp}$ A333 (Fig. 1d).

We anticipated that interacting residues would utilize highly conserved side chain atoms to mediate the formation of the AlgKX$_{Pp}$ complex. However, the majority of the interactions made by the N-terminus of AlgX$_{Pp}$ use main chain atoms and four of the nine AlgX$_{Pp}$ interaction residues are not conserved (Fig. 1e). Thus, we questioned how AlgKX interaction specificity is conferred. To gain insight into this, we used the AlphaFold2 AI program[27] to generate models of the AlgKX complex from *Pseudomonas syringae* (AlgKX$_{Ps}$) and *P. aeruginosa* (AlgKX$_{Pa}$) (Supplementary Fig. 5a). The AlgKX$_{Ps}$ and AlgKX$_{Pa}$ models strongly resemble the AlgKX$_{Pp}$ crystal structure with C$_\alpha$ RMSDs of 0.847 and 0.681 Å, respectively. In both models, the same interaction interface observed in the AlgKX$_{Pp}$ crystal structure was predicted, with the N-terminus of AlgX using only main chain atoms to mediate its interaction with AlgK (Supplementary Fig. 5b, c). To understand how specificity might be conferred, we next examined the surface hydrophobicity of AlgKX$_{Pp}$, AlgKX$_{Pa}$, and AlgKX$_{Ps}$. This analysis revealed a highly conserved, hydrophobic patch on the N-terminus of AlgX that is buried in a conserved, deep hydrophobic groove on AlgK (Supplementary Fig. 6a, b). The hydrophobic groove on AlgK is predominantly composed of isoleucine, leucine, and valine, resides that have been previously established to form hydrophobic clusters and mediate protein–protein interactions[28] (Supplementary Fig. 6c). As this conserved hydrophobic interaction is observed across all three complexes, we propose that this hydrophobic interaction is responsible, at least in part, for conferring interaction specificity.

Examination of the three models also revealed that residues outside of the N-terminus of AlgX use side chain atoms in their interaction with AlgK across all three species, suggesting that additional specificity may be conferred by interactions between the two proteins outside the N-terminus of AlgX. For example, AlgKX$_{Pa}$ and AlgKX$_{Pp}$ each have at least one side chain-side chain interaction occurring outside the N-terminus of AlgX with AlgK (Fig. 1d, e).

In AlgKX$_{Pa}$, we find that the side chains of E166 and Y237 interact with the side chains of AlgK$_{Pa}$ R331 and R357, respectively (Supplementary Fig. 5c, d, e). This interaction is unique in *P. aeruginosa* and not observed in the other two species. Additionally, the side chains of AlgX$_{Pa}$ K172 and R211 interact with the main chain on AlgK$_{Pa}$ G361 and Q328, respectively. In AlgKX$_{Pp}$, the side chain of AlgX$_{Pp}$ R174 interacts with the side chain of AlgK$_{Pp}$ N368 and the main chain of AlgK$_{Pp}$ G367, while the side chain of AlgX$_{Pp}$ K213 interacts with the main chain of AlgK$_{Pp}$ A333 (Fig. 1d). The corresponding residues in AlgX$_{Ps}$ are R172 and I211. While R172 uses its side chain atoms to interact with the main chain of AlgK$_{Ps}$ G358, I211 is not involved in the interaction (Supplementary Fig. 5b, d). Instead, a compensatory interaction occurs with a downstream residue, with the side chain of R214 interacting with the main chain of AlgK$_{Ps}$ Y318. As no side chain-side chain interactions are observed in the AlgKX$_{Ps}$ model, conferred specificity by this additional mechanism is not uniform across all species of AlgKX.

### The N-terminus of AlgX is required for complex formation with AlgK
To validate the identified AlgKX$_{Pp}$ interaction interface, we investigated the importance of the N-terminus of AlgX$_{Pp}$ in complex formation. First, we examined the complex formation between AlgK$_{Pp}$ and AlgX$_{Pp}$ by co-elution using size-exclusion chromatography. The retention volumes of purified AlgK$_{Pp}$ or AlgX$_{Pp}$ were determined by analytical gel filtration and their apparent molecular weights (MW) were calculated by interpolation from a standard curve. AlgX$_{Pp}$ eluted

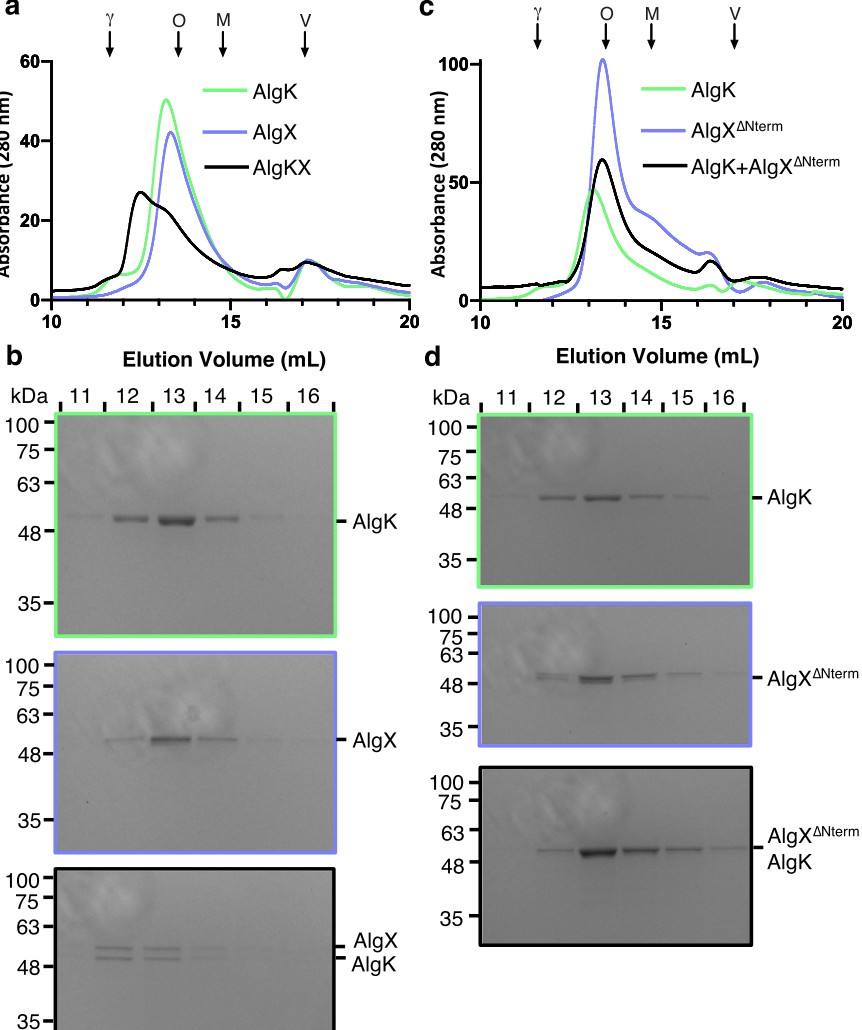

**Fig. 2 | The N-terminus of AlgX is required for interaction with AlgK. a** Gel filtration chromatograms of AlgK$_{Pp}$ (light green), AlgX$_{Pp}$ (periwinkle), and the AlgKX$_{Pp}$ complex (black) overlaid. This experiment was repeated independently one other time. **b** Coomassie-stained SDS-polyacrylamide gels corresponding to the indicated fractions (top) from the experiments in panel **a**. Gel images are outlined using the same color scheme as in panel **a**. This experiment was not repeated. **c** Gel filtration chromatograms of AlgK$_{Pp}$ (light green), AlgX$_{Pp}$$^{\Delta Nterm}$ (periwinkle), and the two added together (black) overlaid. This experiment was repeated independently one other time. **d** Coomassie-stained SDS-polyacrylamide gel corresponding to the indicated fractions (top) from the experiments in panel **c**. Gel images are outlined using the same color scheme as in panel **c**. This experiment was not repeated. Molecular weight standards are indicated by the arrows: γ γ-globulin, O ovalbumin, M myoglobin, and V vitamin B12 with molecular weights 158, 44, 17, and 1.35 kDa, respectively. Source data are provided as a Source Data file.

at a calculated MW of 59.6 kDa compared with its expected MW of 53.4 kDa (Fig. 2a). AlgK$_{Pp}$ eluted at a calculated MW of 65.1 kDa, which is larger than its expected MW of 49.1 kDa (Fig. 2a). This is most likely due to an increased Stokes' radius that results from the elongated nature of TPR domains, as was previously observed with the TPR-containing protein PelB[29]. The AlgKX$_{Pp}$ complex was found to co-migrate and elute at a calculated MW of 123.6 kDa, which is approximately the combined calculated MWs of AlgK$_{Pp}$ and AlgX$_{Pp}$ at a 1:1 molar ratio (Fig. 2a). SDS-PAGE analysis confirmed the co-elution of AlgK$_{Pp}$ and AlgX$_{Pp}$ (Fig. 2b).

Using the same co-elution assay, we next investigated in vitro complex formation with AlgK$_{Pp}$ and a mutant of AlgX$_{Pp}$ that lacks the first 38 residues of its N-terminus (AlgX$_{Pp}$$^{\Delta Nterm}$). In this experiment, AlgK$_{Pp}$ eluted at a calculated MW of 69.9 kDa, while AlgX$_{Pp}$$^{\Delta Nterm}$ eluted at a calculated MW of 55.5 kDa compared with its expected MW of 51.6 kDa (Fig. 2c). When AlgK$_{Pp}$ and AlgX$_{Pp}$$^{\Delta Nterm}$ were combined in a 1:1 molar ratio, the peak eluted at a calculated MW of 55.5 kDa, indicating that the N-terminus of AlgX$_{Pp}$ is required for complex formation with AlgK$_{Pp}$ (Fig. 2c, d).

## The AlgKX complex directly binds alginate

AlgX is essential for alginate acetylation in vivo in *P. aeruginosa* and has been shown to directly bind to and acetylate mannuronic acid oligosaccharides in vitro[18,19,30]. Examining the surface electrostatic properties of AlgKX$_{Pp}$, we discovered an electropositive region spanning 44 Å across the surface of AlgKX$_{Pp}$ that extends from the active site of AlgX$_{Pp}$ to the concave surface of AlgK$_{Pp}$, from R9 towards the N-terminus (Fig. 3a). This electropositive region has potential binding sites for the negatively charged alginate polymer and suggests that AlgK$_{Pp}$ might be involved in guiding export of alginate once it has been acetylated by AlgX$_{Pp}$. To investigate the ability of AlgK$_{Pp}$ and AlgKX$_{Pp}$ to bind alginate in vitro, we used a direct ESI-MS binding assay with oligosaccharides of defined lengths composed of either ManA residues (polyM), or both ManA and GulA residues (polyMG) (Supplementary Figs. 7, 8).

We were able to detect the AlgKX$_{Pp}$ complex and calculate an association constant ($K_a$) of $8.9 \pm 0.9 \times 10^4 \, M^{-1}$ for a 1:1 ratio of complex formation (Supplementary Fig. 9c, d). Representative ESI mass spectra of AlgK$_{Pp}$ and AlgX$_{Pp}$ are shown in Supplementary Fig. 9a, b. We next

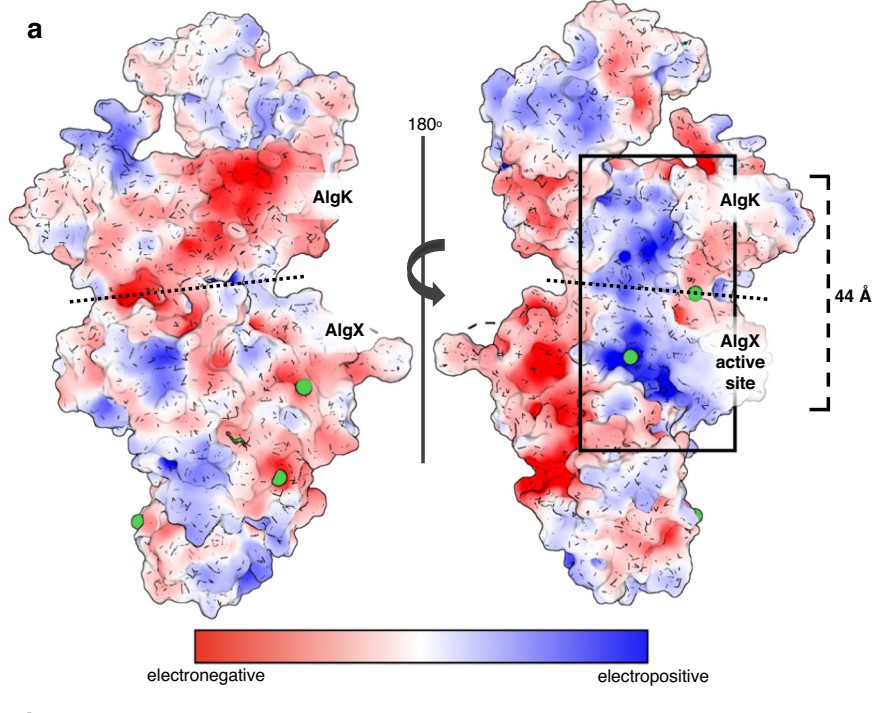

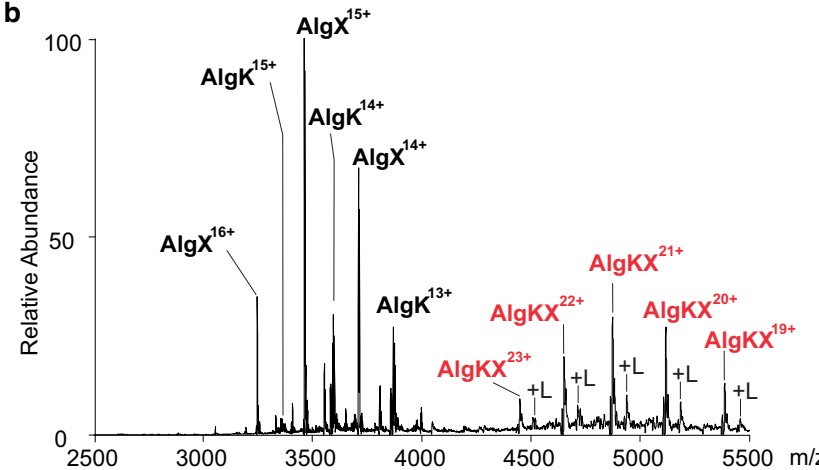

**Fig. 3 | The AlgKX complex binds alginate. a** Electrostatic surface representation of the AlgKX$_{Pp}$ complex calculated by APBS Tools; contoured from +5 (blue) to −5 (red) kT/e. The box highlights the proposed pathway from the site of alginate acetylation by AlgX$_{Pp}$ towards alginate export via AlgK$_{Pp}$. The dashed bracket represents the distance of the proposed pathway. Dotted black lines represent the boundary between AlgK$_{Pp}$ and AlgX$_{Pp}$. Nickel (filled green circles) can be observed. **b** Representative ESI mass spectrum acquired for ammonium acetate (200 mM, pH 7.0) solutions of AlgK$_{Pp}$ and AlgX$_{Pp}$ (5 μM each protein) with GMGMGMG ligand (100 μM). AlgK$_{Pp}$ ≡ AlgK, AlgX$_{Pp}$ ≡ AlgX, AlgKX$_{Pp}$ ≡ AlgKX. Ion signals specific for AlgKX$_{Pp}$ complex with ligand bound are indicated by +L.

assessed the ability of AlgK$_{Pp}$, AlgX$_{Pp}$, and AlgKX$_{Pp}$ to bind alginate ligands of defined lengths and composition and determined their $K_a$ (Table 1 and Supplementary Fig. 10). As expected, we were able to detect the binding of polyM substrates by AlgX$_{Pp}$. AlgX$_{Pa}$ was previously shown to bind longer polyM ligands with increasing $K_a$[19], however, we did not observe this trend with AlgX$_{Pp}$. We also found that GMGMGM and GMGMGMG bound to AlgX$_{Pp}$ with $K_a$'s of $4.5 \pm 1.0$ and $2.7 \pm 0.3 \times 10^2$ M$^{-1}$, respectively. Although *P. aeruginosa* does not make alginate with consecutive GulA residues[27], and therefore GMGGMG is not a biologically relevant ligand, AlgX$_{Pp}$ was able to bind GMGGMG with a $K_a$ of $1.4 \pm 0.3 \times 10^2$ M$^{-1}$. TPR-containing proteins are not generally known for their oligosaccharide binding capabilities, however, we determined that, with the exception of GMGMGMG, AlgK$_{Pp}$ is capable of directly binding polyM and polyMG ligands with similar affinities as AlgX$_{Pp}$. Neither AlgX$_{Pp}$ nor AlgK$_{Pp}$ demonstrated a clear preference for binding polyM or polyMG (Table 1). Strikingly, we were

able to detect the binding of the AlgKX$_{Pp}$ complex to both polyM and polyMG ligands (Table 1 and Supplemental Fig. 10), demonstrating that a polymer-modifying enzyme-TPR-containing protein complex directly interacts with its relevant exopolysaccharide. Furthermore, for some of the ligands tested, including the biologically relevant GMGMGM and GMGMGMG ligands, AlgKX$_{Pp}$ demonstrates greater binding compared to AlgX$_{Pp}$ or AlgK$_{Pp}$ alone, suggesting that the complex binds more tightly to the alginate polymer. A representative ESI mass spectrum acquired for the aqueous ammonium acetate solution of AlgKX$_{Pp}$ and GMGMGMG is shown in Fig. 3b. No ligand binding was detected with AlgKX$_{Pp}$ and pentadeca-hyaluronic acid (HA$_{15}$) (Supplementary Fig. 11).

TPR-containing proteins have previously been demonstrated to influence the activity of polymer-modifying enzymes in vitro in the Pel biosynthetic system[29]. Thus, we next assessed whether the presence of AlgK$_{Pp}$ influences the in vitro enzymatic activity of AlgX$_{Pp}$. Initially, we

**Table 1 | Apparent association constants ($K_a$) for AlgK$_{Pp}$, AlgX$_{Pp}$, and AlgKX$_{Pp}$ for short polymannuronic and polymannuronic-guluronic oligosaccharides as determined by direct ESI-MS in 200 mM aqueous ammonium acetate solution at pH 7 and 25 °C. Source data are provided as a Source Data file**

| Ligand Name | $K_a$ (M$^{-1}$) (AlgK$_{Pp}$)[a] | $K_a$ (M$^{-1}$) (AlgX$_{Pp}$)[a] | $K_a$ (M$^{-1}$) (AlgKX$_{Pp}$)[a] |
|---|---|---|---|
| ManA$_6$ | $(3.1 \pm 0.5) \times 10^2$ | $(1.9 \pm 0.7) \times 10^2$ | $(7.8 \pm 0.8) \times 10^2$ |
| ManA$_7$ | $(4.2 \pm 0.4) \times 10^2$ | $(2.6 \pm 0.3) \times 10^2$ | $(10 \pm 1.0) \times 10^2$ |
| ManA$_{10}$ | $(2.6 \pm 1.0) \times 10^2$ | $(2.4 \pm 1.0) \times 10^2$ | $(5.9 \pm 0.2) \times 10^2$ |
| ManA$_{11}$ | $(1.7 \pm 0.4) \times 10^2$ | $(2.0 \pm 1.0) \times 10^2$ | $(9.3 \pm 0.4) \times 10^2$ |
| ManA$_{12}$ | $(3.5 \pm 0.8) \times 10^2$ | $(2.6 \pm 1.0) \times 10^2$ | $(4.3 \pm 0.2) \times 10^2$ |
| GMGMGM | $(2.5 \pm 0.1) \times 10^2$ | $(4.5 \pm 1.0) \times 10^2$ | $(12 \pm 0.7) \times 10^2$ |
| GMGMGMG | nb | $(2.7 \pm 0.3) \times 10^2$ | $(18 \pm 0.6) \times 10^2$ |
| GMGGMG | $(3.0 \pm 0.9) \times 10^2$ | $(1.4 \pm 0.3) \times 10^2$ | $(5.4 \pm 0.5) \times 10^2$ |

[a]Errors correspond to one standard deviation; nb indicates no binding detected.

monitored the rate of acetylesterase activity—the ability to remove acetyl groups—of AlgK$_{Pp}$, AlgX$_{Pp}$, and AlgKX$_{Pp}$ using 4-nitrophenyl acetate as an acetyl group donor, with the removal of the acetate being monitored at 405 nm in real-time. Although it appears that the AlgK$_{Pp}$-AlgX$_{Pp}$ interaction increased acetylesterase activity compared to AlgX$_{Pp}$ alone, we believe this is an additive effect due to the unexpectedly high background of AlgK$_{Pp}$ and, therefore, not a true increase in activity as a result of complex formation (Supplementary Fig. 12a). We next assessed whether the presence of either polyM or polyMG—the acetyl group acceptor—influenced acetylesterase activity. We baseline-corrected values against the polyM (Supplementary Fig. 12b) and polyMG (Supplementary Fig. 12c) data. The addition of polyM to either AlgK$_{Pp}$ or AlgX$_{Pp}$ did not significantly increase activity, while addition to AlgKX$_{Pp}$ resulted in a significant increase in acetylesterase activity (Supplementary Fig. 12b). Furthermore, the addition of

polyMG to AlgK$_{Pp}$, AlgX$_{Pp}$, and AlgKX$_{Pp}$ significantly increased activity (Supplementary Fig. 12c). As AlgK$_{Pp}$ is not an acetyltransferase enzyme, the observed increase is most likely due to nonspecific hydrolysis of the pseudosubstrate. Overall, the data demonstrate that the addition of an acetyl group acceptor, either polyM or polyMG, influences AlgKX$_{Pp}$ acetylesterase activity.

## AlgK does not interact with other alginate-modifying enzymes

We have demonstrated that AlgK$_{Pp}$ forms a complex with the polymer-modifying enzyme AlgX$_{Pp}$. Previously, it was suggested that AlgK, AlgX, and the alginate epimerase AlgG form a periplasmic complex that guides the polymer for export[31–34]. It is currently unclear how AlgG associates with the rest of the alginate biosynthetic complex to modify the polymer. Nonetheless, we hypothesized that AlgK might act as a scaffold protein to recruit enzymes that directly modify the polymer prior to export. Prior to investigating whether AlgK interacts with AlgG, we first performed in vivo co-immunoprecipitations (co-IP) with vesicular stomatitis virus glycoprotein (VSV-G) tagged proteins complemented into our PAO1 Δ*wspF* P$_{BAD}$*alg* Δ*algK* and PAO1 Δ*wspF* P$_{BAD}$*alg* Δ*algX* strains to validate that our co-IP assay can detect the formation of the AlgKX$_{Pa}$ complex. As we have shown previously, our PAO1 Δ*wspF* P$_{BAD}$*alg* (parental) strain allows for the induction of alginate biosynthetic protein expression using L-arabinose in a high c-di-GMP background[25]. Complementation was performed through integration at the chromosomal *attTn7* site with the complemented gene under the control of an L-arabinose-inducible promoter[25]. As anticipated, VSV-G-tagged AlgK$_{Pa}$ and VSV-G-tagged AlgX$_{Pa}$ were able to pull down AlgX$_{Pa}$, and AlgK$_{Pa}$, respectively (Fig. 4a). Having validated the assay, we proceeded to complement our PAO1 Δ*wspF* P$_{BAD}$*alg* Δ*algG* strain with VSV-G-tagged AlgG$_{Pa}$. Western blot analysis of the co-IP elution sample revealed that AlgG$_{Pa}$ does not pull down AlgK$_{Pa}$ (Fig. 4b). To determine whether the AlgK$_{Pa}$-AlgG$_{Pa}$ interaction requires the presence of AlgX$_{Pa}$ or a preassembled AlgKX$_{Pa}$ complex, we confirmed expression of AlgX$_{Pa}$ in our PAO1 Δ*wspF* P$_{BAD}$*alg* Δ*algG* strain with VSV-G-tagged AlgG$_{Pa}$ strain (Fig. 4c and Supplementary Fig. 13).

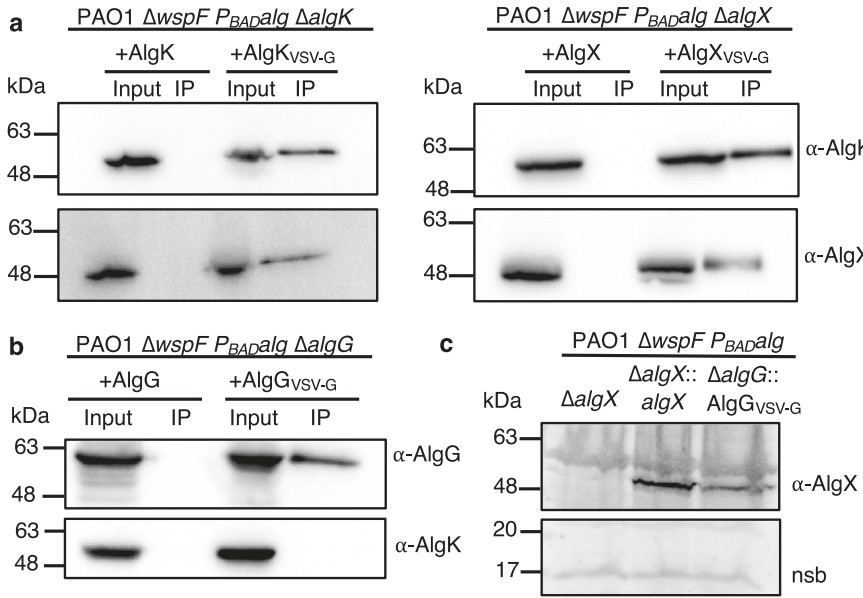

**Fig. 4 | AlgK and AlgG do not interact in vivo in *P. aeruginosa*. a** Co-immunoprecipitation studies from whole cell lysates with untagged and VSV-G-tagged AlgK$_{Pa}$ (left), and untagged and VSV-G-tagged AlgX$_{Pa}$ (right) as the bait. This experiment was repeated independently one other time. **b** Co-immunoprecipitation studies from whole cell lysates with untagged and VSV-G-tagged AlgG$_{Pa}$ as the bait. Proteins applied to the α-VSV-G co-IP resin (input) and the proteins bound to the resin after washing (IP) were analyzed by Western blotting

using alginate-protein-specific antibodies, as indicated. This experiment was not repeated. **c** Western blot analysis of whole cells indicates that AlgX$_{Pa}$ is expressed in the PAO1 Δ*wspF* P$_{BAD}$*alg* Δ*algG*::AlgG$_{VSV-G}$ strain. nsb indicates a nonspecific band in the Western blot analysis that acts as a loading control. Ponceau S staining on the membrane to indicate total protein loading in each well is shown in Supplementary Fig. 12. This experiment was repeated independently one other time. Source data are provided as a Source Data file.

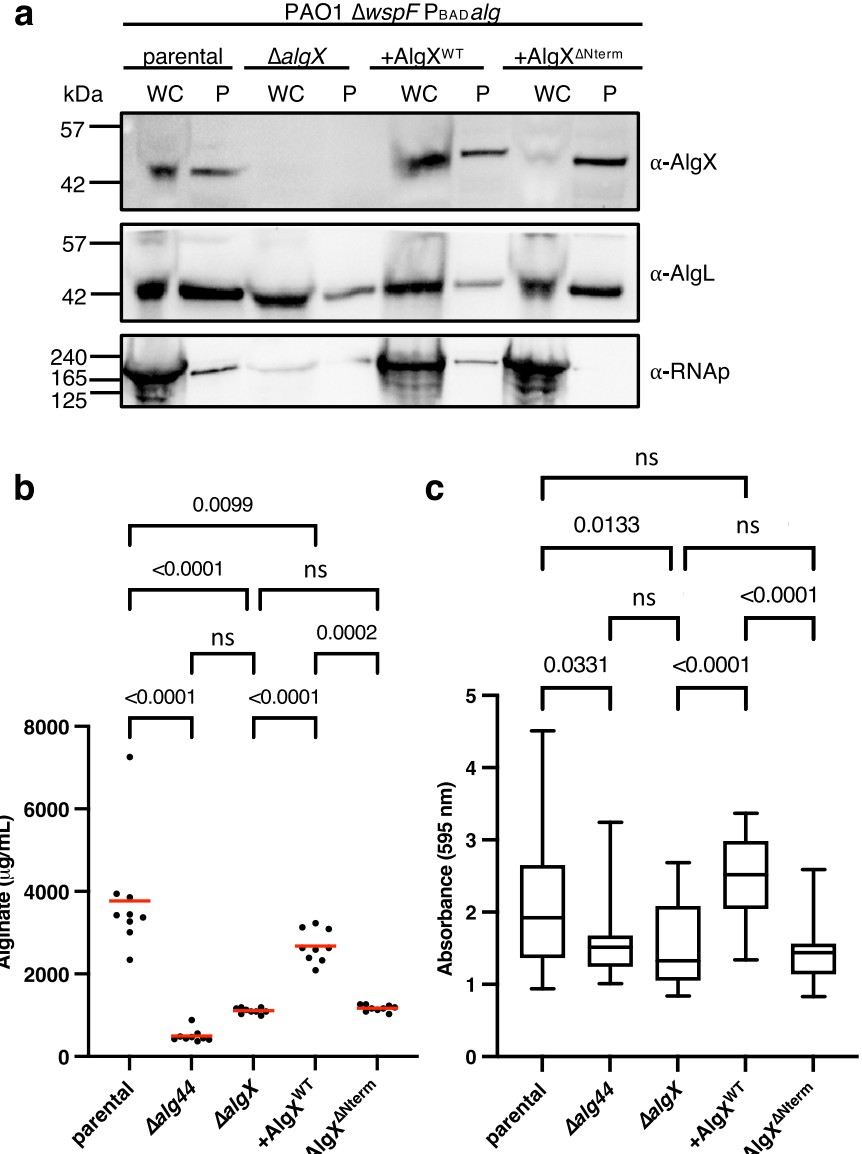

**Fig. 5 | Formation of the AlgKX complex contributes to alginate production and biofilm adherence in *P. aeruginosa*. a** Western blot analysis of the normalized whole cell sample (WC) used for the periplasmic extraction and normalized periplasmic fractions (P) of PAO1 Δ*wspF* P$_{BAD}$ *alg* (parental), PAO1 Δ*wspF* P$_{BAD}$ *alg* Δ*algX* (Δ*algX*), PAO1 Δ*wspF* P$_{BAD}$ *alg* Δ*algX attTn7*::P$_{BAD}$*algX* (+AlgX$^{WT}$), and PAO1 Δ*wspF* P$_{BAD}$ *alg* Δ*algX attTn7*::P$_{BAD}$AlgX$^{ΔNterm}$ (+AlgX$^{ΔNterm}$) strains expressing AlgX, AlgL (periplasmic control and loading control for P), and RNA polymerase (RNAp–cytoplasmic control). Detection of AlgX, AlgL, and RNAp was performed on the same blot. This experiment was repeated independently two other times. **b** Quantification of alginate produced over the course of 24 h by the indicated strains with the addition of 0.5% (w/v) L-arabinose to the growth media. Values represent three technical replicates across three separate experiments ($n = 9$). Red lines represent the mean. **c** Crystal violet staining assay to quantify adherent biofilm biomass. Values represent all technical replicates across three separate experiments ($n = 26$). Whiskers represent the min and max, the center line in the box represents the median, and the hinges of the box represent the 25th and 75th percentiles. Statistical analysis was carried out using a one-way analysis of variance with Bonferroni correction: ns indicates not significant, *p* values indicated directly on graphs. Source data are provided as a Source Data file.

Using the same approach, we have previously demonstrated that the periplasmic lyase, AlgL$_{Pa}$, does not associate with AlgK$_{Pa}$[25].

### Formation of the AlgKX complex is required for alginate production in *P. aeruginosa*

Previous studies reported that deletion of either *algK* or *algX* compromised alginate secretion in *P. aeruginosa*[32,35,36]. To investigate the role of the AlgK$_{Pa}$-AlgX$_{Pa}$ interaction in vivo and its impact on alginate production, we generated a AlgX$_{Pa}$$^{ΔNterm}$ construct that retains its native signal sequence for localization to the periplasm to complement our *P. aeruginosa* PAO1 Δ*wspF* P$_{BAD}$*alg* Δ*algX* strain (Δ*algX*). First, we confirmed the expression and transport of AlgX$_{Pa}$$^{ΔNterm}$ to the periplasm via periplasmic extraction. To directly compare protein expression levels across all strains, we normalized the optical density of the cell culture at 600 nm (OD$_{600}$) for whole cell inputs and the absorbance at 280 nm, representing the total protein content, of periplasmic fractions (Fig. 5a). Although PAO1 Δ*wspF* P$_{BAD}$*alg* Δ*algX attTn7*::P$_{BAD}$AlgX$^{ΔNterm}$ (+AlgX$^{ΔNterm}$) appears to express less AlgX$_{Pa}$ in the whole cell inputs compared to the parental and PAO1 Δ*wspF* P$_{BAD}$*alg* Δ*algX attTn7*::P$_{BAD}$*algX* (+AlgX$^{WT}$), AlgX$_{Pa}$ is detected at comparable levels in periplasmic fractions in the parental, +AlgX$^{WT}$, and +AlgX$^{ΔNterm}$ strains (Fig. 5a). As anticipated, AlgX$_{Pa}$ was not detected in Δ*algX*. AlgL$_{Pa}$ acts as a loading control for the periplasm, while RNA polymerase (RNAp) is a cytoplasmic control.

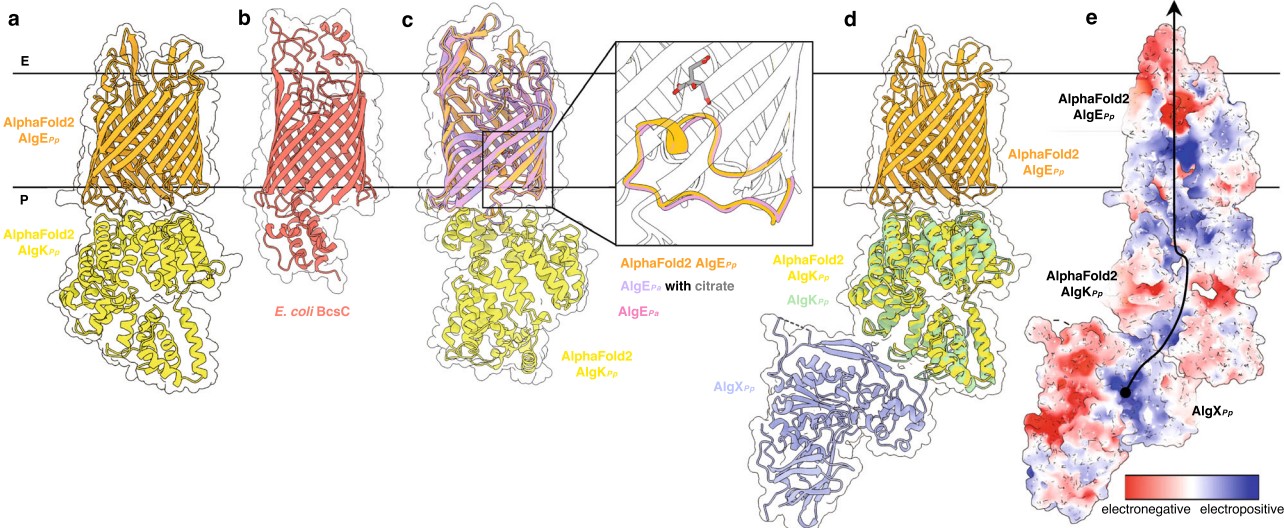

**Fig. 6 | Model of the AlgEKX outer membrane alginate modification and secretion complex. a** AlphaFold2 model of the *P. putida* AlgEK complex (AlgE$_{Pp}$, orange; AlgK$_{Pp}$, yellow). **b** *E. coli* BcsC structure (PDB: 6TZK)[40] (salmon). **c** Superimposition of the AlphaFold2 AlgE$_{Pp}$ (orange) with AlgE$_{Pa}$ (pink, PDB: 3RBH)[33] and AlgE$_{Pa}$ (light purple) complexed with citrate (gray, PDB: 4AFK)[22], which acts as a mimic of mannuronate/guluronate. The inset highlights the AlgE periplasm-facing T8 loop in a closed state. **d** Model of the outer membrane AlgEKX$_{Pp}$ complex. The AlphaFold2 AlgK$_{Pp}$ model (yellow) is superimposed with AlgK from the *P. putida* AlgKX structure (AlgK$_{Pp}$, light green; AlgX$_{Pp}$, periwinkle). **e** Electrostatic surface representation of the AlgEKX$_{Pp}$ complex calculated by APBS Tools; contoured from +5 (blue) to −5 (red) kT/e. Residues 126–190 of AlphaFold2 AlgK$_{Pp}$ and 9–18, 31–41, 61–81, and 356–467 of Alphafold2 AlgE$_{Pp}$ are deleted to show the electropositive pore region. Black arrow indicates the proposed trajectory of polymer export. E extracellular, P periplasm.

We next examined the amount of alginate secreted by Δ*algX*, +AlgX$^{WT}$, and +AlgX$^{ΔNterm}$. As anticipated, Δ*alg44*, which lacks the c-di-GMP binding protein required to initiate alginate production, and Δ*algX* produce significantly less alginate than the parental strain (Fig. 5b). As we have previously shown that AlgX activity is required for alginate acetylation but not for alginate secretion in *P. aeruginosa*[18], the loss of alginate secretion we observe in Δ*algX* is most likely due to impaired alginate export and subsequent degradation by AlgL rather than loss of AlgX enzymatic activity. Complementation of *algX* in the Δ*algX* strain (+AlgX$^{WT}$) successfully reconstituted alginate production, although less is produced compared to the parental strain. +AlgX$^{ΔNterm}$ produced significantly less alginate compared to +AlgX$^{WT}$ and no significant difference was observed when compared to Δ*algX*. We observed similar trends across strains when we assessed biofilm adherence using the crystal violet microtitre dish adherence assay (Fig. 5c). Δ*alg44* and Δ*algX* resulted in significantly less crystal violet staining than the parental, as indicated by the reduced absorbance at 595 nm. +AlgX$^{WT}$ had comparable staining to the parental. +AlgX$^{ΔNterm}$ resulted in significantly less staining compared to +AlgX$^{WT}$, and no significant difference was observed when compared to Δ*algX* (Fig. 5c). To confirm that the observed reduction in biofilm adherence is not due to AlgX$^{ΔNterm}$ enzyme inactivity, we conducted an acetylesterase activity assay and demonstrate that AlgX$^{ΔNterm}$ and AlgX have comparable in vitro enzymatic activity (Supplementary Fig. 14). Overall, our data demonstrate that the AlgK$_{Pp}$-AlgX$_{Pp}$ interaction is required for *P. aeruginosa* alginate production and influences biofilm attachment.

### Structural model of the AlgEKX modification and secretion complex

Previous studies have shown that AlgK$_{Pa}$ contributes to the proper localization of the outer membrane porin AlgE$_{Pa}$[23] and that AlgK$_{Pa}$ co-elutes with a FLAG-tagged AlgE$_{Pa}$[36]. Comparison to other synthase-dependent secretion systems also supports that AlgE$_{Pa}$ and AlgK$_{Pa}$ form a complex, as PelB and PgaA involved in Pel and PNAG biosynthesis and secretion, respectively, each contain both a porin and a TPR domain[37,38]. The ability of AlgK$_{Pp}$ and AlgKX$_{Pp}$ to bind alginate ligands further implies that AlgKX$_{Pa}$ functions in conjunction with AlgE$_{Pa}$ to export the polymer after modification. Thus, we sought to generate a structural model that illustrates how the processes of alginate acetylation and export are coordinated in *Pseudomonas*. While individual structures of AlgE$_{Pa}$[39] and AlgK$_{Pf}$[23] are available, no experimentally determined structure of the complex is currently available. Therefore, we used the AlphaFold2 AI program[27] to generate a model of the AlgEK$_{Pp}$ secretion complex[34]. Due to the low confidence score in the modeled AlgK$_{Pp}$ C-terminus (residues 447–484) (Supplementary Fig. 15), we removed the C-terminus from our final model (Fig. 6a).

The predicted AlgEK$_{Pp}$ structure resembles *Escherichia coli* BscC, the β-barrel porin and TPR-domain containing protein in the cellulose system (Fig. 6b)[40]. The structure of the porin of BscC and its terminal TPR provides precedence for the predicted orientation of AlgK in the AlgEK$_{Pp}$ complex. Furthermore, molecular dynamics (MD) simulations with AlgE$_{Pa}$ and AlgK$_{Pf}$ oriented AlgEK in a similar end-to-end pose[22]. A periplasm-facing loop in AlgE$_{Pa}$ (residues 438–455), termed the T8 loop, was shown to adopt a closed position and protrude into the porin (Fig. 6c)[24]. Interaction with AlgK$_{Pa}$ was proposed to cause a conformational change in the T8 loop to an open position and enable alginate export. One limitation of AlphaFold2 is that it is unable to predict conformational changes, therefore we were not surprised that, in AlgEK$_{Pp}$, the T8 loop remained in a closed position (Fig. 6c). How AlgE transitions to the open conformation remains an unresolved question, but interestingly, a similar periplasmic-facing loop is not observed in BscC[40].

To investigate how alginate acetylation and export are coupled, we superimposed the AlgKs from the Alphafold2-predicted AlgEK$_{Pp}$ model and the AlgKX$_{Pp}$ crystal structure to generate a model of the AlgEKX$_{Pp}$ modification and secretion complex (Fig. 6d). Although the order in which alginate is modified by acetylation by the terminal acetyltransferase AlgX and epimerization by AlgG is still ambiguous, our model suggests that polymer acetylation immediately precedes export. Most noticeably, in the AlgEKX$_{Pp}$ model, AlgK$_{Pp}$ is positioned on AlgE$_{Pp}$ in an orientation which creates an electropositive pore from the active site of AlgX$_{Pp}$ to the extracellular-facing pore of AlgE$_{Pp}$, suggesting a path for alginate export after modification by acetylation (Fig. 6e). Overall, our model provides structural insight into how

alginate biosynthetic proteins in the outer membrane and periplasm coordinate their interaction with each other for effective polymer production.

## Discussion

In this study, we characterized the interaction between the periplasmic alginate-modifying enzyme AlgX and the TPR-containing protein AlgK. We determined the structure of the AlgKX complex from *P. putida* to 2.5 Å, revealing an interaction interface composed primarily of the N-terminus of AlgX$_{Pp}$ and TPRs 9–10 of AlgK$_{Pp}$ (Fig. 1). Using mutagenesis in conjunction with size-exclusion chromatography, we confirmed that the N-terminus of AlgX$_{Pp}$ is required for interaction with AlgK$_{Pp}$ (Fig. 2) and demonstrated using ESI-MS that AlgKX$_{Pp}$ binds polyM and polyMG ligands (Table 1). Assessing the importance of the AlgKX$_{Pp}$ complex in *P. aeruginosa*, we showed that loss of the AlgX$_{Pp}$-AlgK$_{Pp}$ interaction results in abrogated alginate production and biofilm attachment, similar to an *algX* deletion (Fig. 5). Lastly, we provided structural insight into alginate biosynthesis at the molecular level by proposing a model for the AlgEKX$_{Pp}$ outer membrane alginate modification and secretion complex (Fig. 6).

Our data demonstrate that AlgX and AlgK form a complex that is required for the successful export of alginate exopolysaccharide in *P. aeruginosa*. Complex formation has been observed in other bacterial polysaccharide biosynthetic systems. For example, in the *P. aeruginosa* Pel biosynthetic system, the interaction between the Pel-modifying enzyme PelA and the multi-domain protein PelB is essential for Pel-dependent biofilm formation[29]. In this system, PelB contains both the TPR domain and an outer membrane porin[29,37]. Similarly, in the *E. coli* PNAG polysaccharide biosynthetic system, the interaction between the β-barrel porin and TPR-domain containing protein PgaA and the PNAG-modifying enzyme PgaB is necessary for biofilm formation[38]. Although studies in PNAG and Pel have previously demonstrated the importance of interactions between polymer-modifying enzymes and TPR-containing proteins, our study elucidates a complex structure of these two components (Fig. 1). Furthermore, we directly demonstrate the ability of our complex to bind exopolysaccharide (Table 1 and Fig. 3). Using an ESI-MS approach, we determined that AlgKX$_{Pp}$ binds both polyM and polyMG ligands, and that presence of polymer significantly increases AlgKX$_{Pp}$ acetylesterase activity (Table 1 and Supplementary Fig. 11).

We show that AlgK does not interact with the alginate epimerase enzyme AlgG (Fig. 4). At the outer membrane, AlgK has been shown to contribute to the proper localization of AlgE[23,34] and AlgK stability is dependent on the presence of AlgX[36]. Importantly, deletion of either AlgI, AlgJ, or AlgF does not affect polymer secretion, although the secreted polymer is not acetylated[30]. Within the acetylation machinery, only deletion of AlgX results in compromised secretion[32], further supporting AlgX's role in guiding alginate to AlgK for export. AlgK has also been shown to directly interact with the inner membrane protein Alg44[41]. This suggests that AlgK may coordinate biosynthetic processes occurring across the inner and outer membranes. To date, only the structure of the cytoplasmic PilZ domain of Alg44 has been determined[16]. Thus, further structural studies will be necessary to elucidate how AlgK interacts with Alg44 and how this interaction may act as a platform across the periplasm to guide the assembly of the biosynthetic complex.

Our AlgEKX$_{Pp}$ complex provides a structural model for how synthase-dependent secretion systems are organized at the outer membrane. Within the periplasm, AlgL does not associate with the rest of the alginate biosynthetic complex[25]. AlgF and AlgJ have been proposed to form an *O*-acetylation reaction center with AlgI[36], but the molecular details of how these proteins interact and how acetyl group transfer is coordinated between AlgIFJ and AlgX prior to polymer modification is unclear, although different models have been proposed[19,41,42].

At the inner membrane, it is established that Alg8 and Alg44 form an active alginate polymerization complex[41]. While structural information for these proteins and the inner membrane PelDEFG complex required for Pel biosynthesis[40] is lacking, insights can be obtained from the *E. coli* cellulose system, as BcsA contains an inner membrane-embedded glycosyltransferase with a PilZ domain that is analogous to Alg8 and Alg44[43]. The Bcs inner membrane complex interacts with two subunits of BcsG, a phosphoethanolamine transferase analogous to AlgX, via BcsA's first transmembrane helix[43]. Similarly, an interaction between AlgX and Alg44 has been demonstrated[36]. In the cytoplasm, BscQ and BscE, involved in polar localization of the cellulose complex and regulation of phosphoethanolamine modification, respectively, interact with the cellulose inner membrane complex via BcsA's PilZ domain[43]. Whether the cytoplasmic enzymes involved in alginate polymer precursor formation, AlgD, AlgA, and AlgC, associate with the inner membrane proteins, how this might occur from a structural perspective, and their network of protein interactions is currently unknown.

As alginate is secreted through AlgE to the extracellular matrix, energy must be produced to drive this process. Currently, it is understood that alginate does not passively diffuse through AlgE and the required energy for transport is provided by the alginate synthesis machinery in the inner membrane. To investigate this, previous MD studies on AlgE and polyMG applied a force to either pull or push the alginate polymer through the pore electropositive of AlgE from the periplasmic to the extracellular side[24]. Tan et al., concluded that AlgE alone does not impart directionality to alginate export, however, alginate export may be facilitated by breathing motions and slight changes in the conformation of the protein[24]. As our AlgEKX$_{Pp}$ model demonstrates, there is an extended electropositive pathway that could guide alginate for export after modification and thus may influence the forces required for alginate export. Similar MD simulations could be conducted using the AlgEK$_{Pp}$ or AlgEKX$_{Pp}$ complex to determine if complex formation imparts directionality on alginate export.

Synthase secretion systems have mainly been studied from a reductionist perspective—understanding how a complex system functions by analyzing its individual components and their broad effects on bacteria and polymer production in vivo. Although there is still much work that can be done at the level of individual proteins, we hope to gain insight into how *P. aeruginosa* coordinates the formation of the alginate biosynthetic complex in its entirety. Future studies tackling this goal will advance our understanding of alginate biosynthesis in *P. aeruginosa*, thus providing insight into potential therapeutic targets designed to abrogate biofilm production and prevent chronic infection in the COVID-19 and CF lung.

## Methods

### Bacterial strains, plasmids, and growth conditions

A complete list of all bacterial strains and plasmids used in this study can be found in Supplementary Table 2. All *P. aeruginosa* strains were derived from PAO1[44]. *P. aeruginosa* mutant and complemented strains were generated using allelic exchange and miniTn7 mutagenesis, as previously described in refs. 45, 46. A list of all primers used in this study can be found in Supplementary Table 3.

Lysogeny broth (LB) contained 10.0 g tryptone, 5.0 g yeast extract, and 5.0 g NaCl per liter of ultrapure water. Vogel-Bonner minimal medium (VBMM) was prepared as a 10× concentrate, which contained 2.0 g MgSO$_4$·7H$_2$O, 20 g citric acid, 100 g K$_2$HPO$_4$, and 35 g NaNH$_4$HPO$_4$·4H$_2$O, per liter of ultrapure water and was adjusted to pH 7.0 and sterilized by filtration. The 10× VBMM solution was diluted as needed. Semisolid media was prepared by adding 1.5% (w/v) agar to LB and VBMM. Where appropriate, antibiotic selection was added to growth media as follows: for *P. aeruginosa*, carbenicillin (Carb) at 300 μg/mL, and gentamicin (Gen) at 30 or 60 μg/mL, depending on

the application as described below; for *E. coli*, Gen at 10 µg/mL, Carb at 100 µg/mL, and kanamycin (Kan) at 50 µg/mL.

## Basic molecular biology methods

Molecular and microbiological techniques were performed according to standard protocols[47]. Genomic DNA (gDNA) isolation, plasmid preparation, and DNA gel extraction were performed using nucleotide purification kits purchased from Bio Basic. All primers were obtained from Sigma-Aldrich.

## Construction of *P. aeruginosa* chromosomal deletions

In-frame and unmarked deletions of *algK*, *algX*, and *algG* in *P. aeruginosa* PAO1 Δ*wspF* $P_{BAD}$*alg* were generated using a previously described protocol[46]. Briefly, flanking upstream and downstream regions of the *algK*, *algX*, and *algG* ORFs were amplified and joined by splicing-by-overlap extension PCR (Supplementary Table 3). Upstream forward and downstream reverse primers were tailed with EcoRI and HindIII restriction enzyme sequences, respectively, to enable cloning of the spliced PCR products. PCR products were gel purified, digested with EcoRI and HindIII (Thermo Fisher Scientific) restriction enzymes as per manufacturer's instructions, and ligated into pEX18Gm[48] using T4 DNA ligase (Thermo Fisher Scientific). The resulting allelic exchange vectors, pEX18Gm::Δ*algK*, pEX18Gm::Δ*algX*, and pEX18Gm::Δ*algG* were selected for on LB agar supplemented with 10 µg/mL Gen, identified by colony PCR, and verified by Sanger sequencing using M13 forward and M13 reverse primers (Supplementary Table 3).

The deletion alleles encoded by pEX18Gm::Δ*algK*, pEX18Gm::Δ*algX*, and pEX18Gm::Δ*algG* were introduced into *P. aeruginosa* PAO1 Δ*wspF* $P_{BAD}$*alg* via bi-parental mating with donor strain *E. coli* SM10[46,49,50]. Merodiploids were selected on VBMM supplemented with 60 µg/mL Gen. SacB-mediated counter-selection was carried out by selecting for double cross-over mutations on no-salt LB (NSLB) agar supplemented with 15% (w/v) sucrose. Unmarked gene deletions were identified by colony PCR with primers flanking the outside regions of *algK*, *algX*, and *algG* (Supplementary Table 3). To confirm each deletion, PCR products were gel purified and sent for Sanger sequencing.

## Construction of miniTn7 vectors

The use of the vector pUC18-mini-Tn7T-Gm[51,52] for single-copy chromosomal integration at the *attTn7* site in *P. aeruginosa* was previously reported[45]. The vector was previously modified for arabinose-dependent expression of complemented genes using the *araC*-$P_{BAD}$ promoter[53] (Supplementary Table 3). The *algK* and *algX* ORFs were amplified using the primer pairs algK_miniTn7_NotI, algK_miniTn7_NcoI and algX_miniTn7_PstI, algX_miniTn7_SacI, respectively, which encode a synthetic ribosome binding site upstream of the start codon (Supplementary Table 3). The resultant PCR products were cloned into pUC18T-miniTn7T-Gm-pBAD using NotI, NcoI, and PstI, SacI restriction enzyme cut sites for *algK* and *algX*, respectively, selected on LB agar with 10 µg/mL Gen and 100 µg/mL Carb, and confirmed by Sanger sequencing using the miniTn7 Seq_F and miniTn7 Seq_R primers (Supplementary Table 3). The AlgX^ΔNterm mutant was generated on the pUCT18T-miniTn7T-AlgX_Pa vector using the ΔN_term_cleavage_site and ΔN_term_2 primer pairs (Supplementary Table 3).

Complemented *P. aeruginosa* strains were generated through the incorporation of miniTn7 vectors at the *attTn7* site on the *P. aeruginosa* chromosome via electroporation of miniTn7 vectors and the pTNS2 helper plasmid, as previously described in ref. 45. Transposon mutants were selected on LB agar supplemented with 30 µg/mL Gen and confirmed by colony PCR using the miniTn7 Seq_F and miniTn7 Seq_R primers (Supplementary Table 3).

## Construction of VSV-G tagged alginate proteins

VSV-G-C-terminally tagged AlgX, AlgK, and AlgG proteins were generated directly on the pUC18-miniTn7-T-AlgX_Pa, pUC18-miniTn7-T-

AlgK_Pa, pUC18T-miniTn7-T-AlgG_Pa vectors, respectively, using primer pairs found in Supplementary Table 3.

## Expression and purification of *P. putida* AlgK and AlgX

The nucleotide sequence of *P. putida* KT2440 AlgK and AlgX were obtained from the *Pseudomonas* Genome Database[54] and codon optimized for expression in *E. coli* by Bio Basic Inc. Codon optimized AlgK^1–484 and AlgX^1–479 were incorporated into the expression plasmids pET26b and pET24b, respectively. The AlgK^33–484 and AlgX^ΔNterm constructs were generated using pET26b *P. putida* AlgK^1–484 and pET24b *P. putida* AlgX^1–479, respectively, as templates (Supplementary Table 2). The plasmids pET26b *P. putida* AlgK^33–484, pET24b *P. putida* AlgX^ΔNterm, and pET24b *P. putida* AlgX^1–479 were transformed into *E. coli* BL21 (DE3) CodonPlus and *E. coli* Lemo21 (DE3) competent cells, respectively. *E. coli* BL21 (DE3) CodonPlus cells were grown in LB with 50 µg/mL Kan at 37 °C and expression was induced with 1 mM isopropyl β-D-1-thio-galactopyranoside (IPTG) when $OD_{600}$ reached 0.7. Induced cells containing either pET26b *P. putida* AlgK^33–484 or pET24b *P. putida* AlgX^ΔNterm were incubated at 37 °C for 4 h. *E. coli* Lemo21 (DE3) cells containing pET24b *P. putida* AlgX^1–479 were grown in LB at 37 °C with 50 ug/mL Kan, 30 ug/mL chloramphenicol to maintain the pLemo plasmid and 100 µM L-rhamnose to inhibit T7 RNA polymerase, thus enabling tunable expression. The expression of *P. putida* AlgX^1–479 was induced with 0.4 mM IPTG when $OD_{600}$ reached 0.7 and induced cells were incubated at 18 °C overnight. For all three C-terminally hexahistidine-tagged proteins, cells were harvested and resuspended in lysis buffer (20 mM Tris-HCl pH 8.0, 500 mM NaCl, 1 mM PMSF, 0.1 mg/mL lysozyme, 0.1 mg/mL DNase, and EDTA-free protease inhibitor cocktail tablet) and lysed by homogenization using an Emulsiflex-C3 (Avestin Inc.) at 10,000 psi. The soluble fraction of the cell lysate obtained by centrifugation was applied to a nickel affinity column pre-equilibrated with Ni-NTA load buffer (20 mM Tris-HCl pH 8.0, 500 mM NaCl, and 10 mM imidazole). The column was washed with 20 mM imidazole to remove non-specifically bound proteins. The His-tagged recombinant proteins were eluted with 300 mM imidazole. The elutions were pooled and concentrated using a 30 KDa cutoff Vivaspin® Turbo 15 centrifugal concentrator and further purified by size-exclusion chromatography using a HiLoad 16/60 Superdex 75 column (GE Healthcare) in 20 mM Tris pH 8.0, 150 mM NaCl, 2% (v/v) glycerol and 50 mM L-Arg/Glu.

## Crystallization, data collection, and refinement

With the goal to crystallize the *P. putida* AlgK-AlgX (AlgKX_Pp) complex, purified AlgK_Pp^33–484 and AlgX_Pp^1–479 were mixed at a 1:1 molar ratio at a concentration of 24 mg/mL and incubated on ice for 30 min. Initial crystallization conditions were screened using the MCSG Suite (Microlytic Crystallization Screens, Anatrace). The crystallization plates were set up using the Gryphon LCP (Art Robbins Instruments) with 1 µL of protein mixture and 1 µL of well solution and incubated at 25 °C in the Rigaku Crystal Minstrel DT crystal imager. After ~90 days, we observed crystals in the initial screen with 1 µL of protein (24 mg/mL in 20 mM Tris pH 8.0, 150 mM NaCl, 2% (v/v) glycerol, and 50 mM L-Arg/Glu) and 1 µL of crystallization solution (0.1 M Tris-HCl pH 8.5, 0.01 M nickel (II) chloride, 20% (w/v) PEG2000 MME). A single crystal was then transferred to the reservoir solution with 20% PEG 400 added as cryoprotectant and flash-frozen in liquid nitrogen for data collection.

X-ray diffraction data were collected on beamline CMCF-BM (08B1-1) at the Canadian Light Source at a wavelength of 1.5120 Å and a temperature of 100 K using MxDC and MxLIVE for remote data collection (Supplementary Table 1). The X-ray data were indexed, integrated, and scaled using XDS[55]. Initial phases were determined using the molecular replacement technique in PHASER[56] with *P. fluorescens* AlgK and *P. aeruginosa* AlgX as the search models (PDB: 3E4B and 4KNC, respectively). The final model of the AlgKX_Pp complex was

generated by iterative rounds of the manual model building using Coot[57] and refinement in PHENIX.REFINE[58]. 22 translation/Libration/Screw groups used during refinement were determined automatically using the TLSMD web server[59]. Ramachandran statistics were calculated by MolProbity[60]: total favored, 97.15 %; total allowed 2.72 %; total outliers, 0.14 %. See Supplementary Table 1 for data statistics.

## Multiple and pairwise sequence alignment

Sequences were taken from UniProt[61]. The UniProt entry identifiers for the sequences are as follows: *P. putida* AlgX (Q88ND0); *P. putida* AlgK (Q88NC7); *P. aeruginosa* AlgX (Q51372); *P. aeruginosa* AlgK (P96956); *P. syringae* AlgX (Q887Q4); *P. syringae* AlgK (Q887Q1). For pairwise sequence alignment, sequences were input into Needle[62]. For multiple sequence alignment, sequences were input into Clustal Omega[62] in FASTA format.

## Analysis tools

The AlgKX binding interface observed in the complex structure was analyzed by PISA[63]. The electrostatic surface potentials were calculated using APBS Tools[64]. Conservation analysis was performed using the ConSurf server[65]. Secondary structure prediction of protein complexes was performed using AlphaFold2 (Alphabet/Google DeepMind)[27]. All structural figures were generated using PyMOL (The PyMOL Molecular Graphics System, Version 1.2r3pre, Schrödinger, LLC) and ChimeraX 1.4 (Resource for Biocomputing Visualization, and Informatics RBVI, UCSF). All column data were visualized and analyzed using GraphPad Prism 9 (Dotmatics).

## Analytical gel filtration

Purified *P. putida* AlgK[33–484], AlgX[1–479], and AlgX[ΔNterm] were applied to a calibrated SEC650 (Bio-Rad) gel filtration column separately to obtain their individual elution profiles. Purified *P. putida* AlgK and AlgX were combined at a 1:1 molar ratio and incubated on ice for 30 min prior to application to the column. The elution profiles were monitored at 280 nm. Apparent molecular weights (MWs) of AlgK, AlgX, and the AlgK-AlgX complex were calculated by interpolating from the MW vs. elution volume standard curve, which was generated by using a Gel Filtration Standard (Bio-Rad) as per the manufacturer's guidelines. Samples were combined with an equal volume of SDS-PAGE sample buffer (4% (w/v) SDS, 0.2% (w/v) bromophenol blue, 20% (v/v) glycerol, and 200 mM dithiothreitol) and boiled at 95 °C for 20 min prior to analysis by SDS-PAGE and Coomassie staining.

## Acetylesterase activity assay

The assay was performed as previously described in refs. 19, 29 with minor modifications. Briefly, standard reactions contained 5 μM protein (AlgK[*Pp*], AlgX[*Pp*], or AlgK[*Pp*]-AlgX[*Pp*] complex) in 50 mM sodium HEPES pH 7.0 and 75 mM NaCl. Reactions were initiated with the addition of *p*NP-acetate dissolved in ethanol to 2.5 mM. Alginate substrates were added at a final concentration of 0.5 mg/mL. Alginate substrates were purified as previously described[62] from FRD1[66] and FRD462[67]. The hydrolysis and removal of an acetate group from *p*NP-acetate was monitored in real-time for 10 min at 405 nm for the appearance of *p*-nitrophenyl. The background hydrolysis rate of *p*NP-acetate in the absence of enzymes was measured and subtracted from reaction rates. For Supplementary Fig. 11b, a similar assay was performed as previously described in ref. 19, with minor modifications. Briefly, Briefly, standard reactions contained 5 μM protein (AlgK[*Pp*], AlgX[*Pp*], or AlgK[*Pp*]-AlgX[*Pp*] complex) in 50 mM sodium HEPES pH 7.0 and 75 mM NaCl. An unacetylated polyM substrate that is ten residues in length (Man10) purchased from Qingdao BZ Oligo Biotech Co. Ltd is added at 1 mg/mL. Reactions were initiated with the addition of 3 mM 3-carboxyumbelliferyl acetate, dissolved in DMSO. The hydrolysis and removal of an acetate group from 3-carboxyumbelliferyl acetate was monitored in real-time for 10 min using an excitation of 386 nm and an emission of 447 nm. The background hydrolysis rate of 3-carboxyumbelliferyl acetate in the absence of enzymes was measured and subtracted from reaction rates. Data were analysed using GraphPad Prism 9 (Dotmatics).

## ESI-MS alginate binding assay

AlgX[*Pp*] and AlgK[*Pp*] stock solutions were buffer exchanged into 200 mM aqueous ammonium acetate (pH 7) using a 10 kDa cut-off Amicon 0.5 mL microconcentrators (EMD, Millipore, Billerica, MA). Stock solutions of oligosaccharides were prepared by dissolving a known amount of solid compound in Milli-Q water and stored at −20 °C until used. Nanoelectrospray (nanoESI) mass spectrometry measurements were performed on a Q Exactive Orbitrap (Orbitrap) mass spectrometer (Thermo Fisher Scientific, Bremen, Germany) and on a Q Exactive Ultra-High Mass Range Orbitrap mass spectrometer (Thermo Fisher Scientific, Bremen, Germany), both equipped with a nanoflow ESI source. NanoESI was performed by applying a voltage of ~0.8 kV to a platinum wire inserted into the nanoESI tip, which was produced from a borosilicate glass capillary (1.0 mm o.d., 0.78 mm i.d.) pulled to ~2 μm o.d. using a P − 1000 micropipette puller (Sutter Instruments, Novato, CA). The inlet capillary of the MS was heated to 120 °C, the S-lens RF level was set at 100, automatic gain control target was set at $1 \times 10^6$ with a maximum injection time of 200 ms. All MS data were acquired and processed using Thermo Xcalibur 4.1 software.

Association constants ($K_{a,kL}$ and $K_{a,xL}$) for interactions of AlgK[*Pp*] and AlgX[*Pp*] with oligosaccharide ligands were obtained using the direct ESI-MS assay[68,69]. The binding measurements were carried out at 21 °C, and the equilibrium mixtures were prepared by mixing aliquots of the stock solutions of proteins and ligands in aqueous ammonium acetate (200 mM). ESI-MS measurements were performed after a 1 h incubation time period. All mass spectra were corrected, when needed, for the occurrence of nonspecific carbohydrate-protein binding during the ESI process using the reference protein method[70]. $K_{a,kL}$ and $K_{a,xL}$ values were determined from the abundance ratio ($R$) of the ligand-bound (PL) to free protein (P) ions (P can be AlgK[*Pp*] or AlgX[*Pp*]), after correction for nonspecific ligand binding, and the initial concentrations of protein ($[P]_0$) and ligand ($[L]_0$), Eq. (1):

$$\frac{R}{R+1} = \frac{K_a[P]_0 + K_a[L]_0 + 1 - \sqrt{(1 - K_a[L]_0 + Ka[P]_0)^2 + 4K_a[L]_0}}{2K_a[P]_0} \quad (1)$$

where $R$ is taken to be equal to the corresponding equilibrium concentration ratio ([PL]/[P]) in solution, Eq. (2):

$$R = \frac{\sum Ab(PL)}{\sum Ab(P)} = \frac{[PL]}{[P]} \quad (2)$$

and $K_a$ is $K_{a,kL}$ or $K_{a,xL}$.

The $K_{a,kx}$ for the interaction of AlgK[*Pp*] and AlgX[*Pp*] with the formation of AlgKX[*Pp*] complex was obtained using the Slow Mixing Mode (SLOMO) ESI-MS assay[71]. Briefly, ~2 μL of a solution of AlgX[*Pp*] (5 μM) and AlgK[*Pp*] (3–10 μM) was introduced into the nanoESI tip, followed by injection of 10 μL of solution of AlgX[*Pp*] (5 μM) and AlgK[*Pp*] (45 μM). The binding measurements were carried out at 21 °C; time-resolved mass spectra were averaged over 1 min intervals and the sum of the charge state-normalized abundances of the AlgK[*Pp*] and AlgKX[*Pp*] complex ions were calculated automatically using the SWARM software (https://github.com/pkitov/CUPRA-SWARM)[72]. The corrected (for response factors) abundances of AlgX[*Pp*] and AlgKX[*Pp*] species were used to calculate $R/(R+1)$ values for AlgK[*Pp*] (designated as L) and AlgX[*Pp*] (designated as P) interaction and association constant $K_{a,kx}$ was found using Eq. (1) ($K_{a,kx}$ in this case designated as $K_a$ in Eq. (1)).

The $K_{a,kxL}$ values for interactions of AlgKX[*Pp*] with oligosaccharide ligands were determined by fitting of the binding model described below (Eqs. 3a–d and 4a–c) to a plot of experimental values

of ligand-bound fraction of AlgKX$_{Pp}$ species, using Maple 2017 (Maplesoft, Waterloo, Canada):

$$AlgK_{Pp}L \rightleftharpoons AlgK_{Pp} + L, \; K_{a,kL} = \frac{[AlgKPpL]}{[AlgKPp][L]} \quad (3a)$$

$$AlgX_{Pp}L \rightleftharpoons AlgX_{Pp} + L, \; K_{a,xL} = \frac{[AlgXPpL]}{[AlgXPp][L]} \quad (3b)$$

$$AlgKX_{Pp} \rightleftharpoons AlgK_{Pp} + AlgX_{Pp}, \; K_{a,kx} = \frac{[AlgKX_{Pp}L]}{[AlgK_{Pp}][AlgX_{Pp}]} \quad (3c)$$

$$AlgKX_{Pp}L \rightleftharpoons AlgKX_{Pp} + L, \; K_{a,kxL} = \frac{[AlgKX_{p}L]}{[AlgKX_{Pp}L][L]} \quad (3d)$$

The corresponding mass balance equations are shown below:

$$[L]_0 = [L] + [AlgK_{Pp}L] + [AlgX_{Pp}L] + [AlgKX_{Pp}L] \quad (4a)$$

$$[AlgK_{Pp}]_0 = [AlgK_{Pp}] + [AlgK_{Pp}L] + [AlgKX_{Pp}] + [AlgKX_{Pp}L] \quad (4b)$$

$$[AlgX_{Pp}]_0 = [AlgX_{Pp}] + [AlgX_{Pp}L] + [AlgKX_{Pp}] + [AlgKX_{Pp}L] \quad (4c)$$

where $[AlgK_{Pp}]_0$, $[AlgK_{Pp}]_0$, and $[L]_0$ are initial concentrations of AlgK$_{Pp}$, AlgX$_{Pp}$, and oligosaccharide ligand L; $K_{a,kL}$, $K_{a,xL}$, and $K_{a,kx}$ are association constant values determined as described above, and $K_{a,kxL}$ is unknown parameter found by the fitting of the model to experimental data. PolyM and polyMG ligands were synthesized as previously described in refs. 73–75.

### Periplasmic extraction

The method of cellular fractionation was adapted and truncated from a previously described protocol[76]. Briefly, 1 L of LB supplemented with 30 μg/mL Gen and 0.5% (w/v) L-arabinose was inoculated with cells from solid media and grown for 16 h at 37 °C with shaking. The OD$_{600}$ was normalized to 1.000 for all samples. A sample of the whole cells was taken for analysis by Western blot. Cells were removed by centrifugation and resuspended in 5 mL 0.2 M Tris-HCl pH 8.0, 1 M sucrose, 1 mM EDTA, and 1 mg/mL lysozyme. Cells were incubated at 21 °C for 5 min prior to the addition of 20 mL ultrapure H$_2$O and incubation on ice for 20 min. The samples were centrifuged at 77,900 × $g$ for 45 min at 4 °C. The supernatant, containing the periplasmic fraction, was concentrated from 25 to 10 mL using a Vivaspin 5000 kDa cut-off concentrator (Sartorius). The total protein concentration for all samples was normalized to an absorbance at 280 nm of 0.1 to allow for comparison of periplasmic protein expression across strains. Samples were taken for analysis by Western blot.

### Purification and quantification of alginate from *P. aeruginosa*

Purification of alginate was carried out as previously described[25]. Briefly, 25 mL of modified alginate-producing defined medium containing 100 mM monosodium glutamate, 7.5 mM monosodium phosphate, 16.8 mM dipotassium phosphate, and 10 mM magnesium sulfate supplemented with 30 μg/mL Gen, with the addition of 0.5% (w/v) L-arabinose, was inoculated with cells from solid media and grown for 22 h at 37 °C shaking. Cells were removed by centrifugation and culture supernatants were collected. To precipitate alginate, 3× volume of cold isopropanol was added to the supernatants and incubated at −20 °C overnight. Precipitated alginates were collected by centrifugation, and excess isopropanol was removed by air drying samples at 21 °C overnight. Samples were collected and resuspended

in 15 mL ultrapure H$_2$O and then lyophilized to dryness using the VirTis BenchTop Pro Freeze Dryer (SP Scientific Products). Samples were resuspended in 1 mL PBS and incubated with 30 μg/mL each DNase I (Bio Basic) and RNase A (Bio Basic) overnight at 37 °C. The following day, samples were incubated with 30 μg/mL proteinase K (Bio Basic) overnight at 37 °C. Samples were dialyzed against ultrapure H$_2$O overnight using a 3.5 kDa molecular weight cutoff dialysis membrane (FisherBrand). Samples were collected and lyophilized to dryness using the VirTis BenchTop Pro Freeze Dryer (SP Scientific Products). Samples were assayed for alginate concentration using a colourimetric test for uronic acids with alginic acid from *Macrocystis pyrifera* (Sigma-Aldrich) used as the standard, as was previously described[77,78]. Briefly, a borate stock solution (4 M H$_3$BO$_3$ in 2 M KOH), borate-sulfuric acid reagent (100 mM H$_3$BO$_3$ in concentrated H$_2$SO$_4$), and carbazole reagent (0.1% (w/v) carbazole in anhydrous ethanol) were made. One mL of borate-sulfuric acid reagent per technical replicate was chilled on ice. Thirty mL of purified alginate or alginate standard was added as a layer on top of the borate-sulfuric acid reagent. Tubes were mixed by vortex for 4 s prior to returning to the ice. Thirty uL carbazole reagent was added and samples were mixed by a vortex. Samples were heated to 55 °C for 30 min and then cooled on ice. Absorbance was measured at 530 nm and the concentration of alginate was calculated from the *M. pyrifera* standard curve.

### Crystal violet microtitre plate assay

*P. aeruginosa* strains were grown to stationary phase in NSLB supplemented with 30 μg/mL Gen and 0.5% (w/v) L-arabinose and were diluted to a final OD$_{600}$ of 0.01 in 1 mL NSLB supplemented with 30 μg/mL Gen and 0.5% (w/v) L-arabinose. About 100 μL of the normalized cultures were added to the wells of a Nunc MaxiSorp flat-bottom 96-well plate (Thermo Fisher) and incubated at 25 °C for 24 h statically. Non-adherent biomass was removed by washing the wells three times with ultrapure water and the remaining biomass was stained by adding 150 μL of 0.1% (w/v) crystal violet for 10 min at 21 °C. Excess stain was removed and wells were washed three times with ultrapure water. 200 μL of anhydrous ethanol was added to each well and incubated for 10 min at 21 °C to solubilize the remaining stain. 40 μL of the solubilized stain was transferred to a separate Nunc plate (Thermo Fisher) with 160 μL anhydrous ethanol (1:5 dilution). The absorbance was measured at 595 nm using an Epoch Microplate Spectrophotometer (BioTek Instruments).

### Co-immunoprecipitation of VSV-G-tagged alginate proteins in P. aeruginosa

Co-immunoprecipitation was carried out as previously described in ref. 25. Briefly, Cells were inoculated into 1 L of LB supplemented with 30 μg/mL Gen and 0.5% (w/v) L-arabinose and grown overnight for 16 h at 37 °C shaking. The following morning, cells were collected by centrifugation at 6700 × $g$ for 30 min at 4 °C. Cell pellets were transferred to a 50 mL conical tube and resuspended in 50 mL lysis buffer (20 mM Tris-HCl pH 8.0, 100 mM NaCl, 1 mM EDTA, 2% (w/v) Triton X-100, 1.0 mg/mL lysozyme, 0.1 mg/mL DNase, and one SIGMA*FAST*™ Protease Inhibitor Cocktail EDTA-free tablet (Sigma-Aldrich). Cells were incubated for 1 h at 4 °C on a rocker. The cell lysates were centrifuged at 20,100 × $g$ for 40 min at 4 °C to remove cellular debris. Anti-VSV-Glycoprotein-Agarose mouse monoclonal antibody beads (Sigma-Aldrich) were resuspended and 60 μL was added to the 50 mL conical tube containing the lysate. The cell lysates were incubated with the agarose beads for 1 h at 4 °C on a rocker. The beads were pelleted by centrifugation at 110 × $g$ for 2 min at 4 °C and the supernatant was carefully decanted. The beads were washed three times with 15 mL lysis buffer without adding DNase or lysozyme. A final wash was done with 15 mL lysis buffer without adding DNase, lysozyme, or Triton X-100. Beads were resuspended in 100 μL 150 mM glycine pH 2.2 and incubated at 21 °C for 15 min to elute the proteins from the agarose beads.

The beads were pelleted by centrifugation at $110 \times g$ for 2 min at 4 °C and the supernatant was carefully transferred to a microfuge tube for storage and 40 μL of 1 M $K_2HPO_4$ was added. Samples were analyzed by Western blot.

## *P. aeruginosa* AlgX gene expression

Overnight cell cultures were grown in LB supplemented with 30 μg/mL Gen and 0.5% (w/v) L-arabinose for 16 h overnight at 37 °C shaking. Cell culture aliquots were normalized to an $OD_{600}$ of 1.000 and centrifuged at $25,000 \times g$ for 10 min to isolate cell pellets. Cell pellets were combined with SDS-PAGE sample buffer (4% (w/v) SDS, 0.2% (w/v) bromophenol blue, 20% (v/v) glycerol, and 200 mM dithiothreitol) in a 1:1 ratio and boiled at 95 °C for 20 min prior to analysis by Western blot.

## Western blot analysis

Samples were combined with an equal volume of SDS-PAGE sample buffer (4% (w/v) SDS, 0.2% (w/v) bromophenol blue, 20% (v/v) glycerol, and 200 mM dithiothreitol) and boiled at 95 °C for 20 min prior to loading each sample onto a 12% (v/v) polyacrylamide gel. Protein was transferred to a polyvinylidene fluoride membrane for immunoblotting. The membrane was blocked using 5% (w/v) skim milk dissolved in TBST (50 mM Tris:HCl pH 7.5, 150 mM NaCl, 0.1% (v/v) Tween-20) for 1 h at 21 °C. Blots were washed twice with TBST and the membrane was then incubated with an $AlgX_{Pa}$ protein-specific polyclonal antibody from rabbit (Cedarlane) at a 1:1000 dilution in TBST and then probed with goat α-rabbit horseradish peroxidase (HRP)-conjugated secondary antibody (Bio-Rad) at 1:300 dilution in TBST for 1 h at 21 °C. Blots were washed five times in TBST. $AlgX_{Pa}$ bands were detected using the Super Signal West Pico chemiluminescent substrate from Pierce (Thermo Scientific). Blots were imaged using the Chemidoc XRS System (Bio-Rad). After detection, the blot was washed three times with water and (for Fig. 5a) stripped using a stripping buffer (50 μM EDTA, 7 M guanidine HCl, 50 mM glycine, 100 mM KCl, 77 μM 2-mercaptoethanol) and incubated for 10 min at 21 °C. The blot was washed three times with TBST. After stripping, the blot was blocked using 5% (w/v) skim milk dissolved in TBST for 1 h at 21 °C. Blots were washed twice with TBST and cut in half. The top half was then incubated with an *E. coli* monoclonal anti-RNA polymerase antibody from a mouse (Invitrogen) at a 1:1000 dilution in TBST and the bottom half was incubated with an $AlgL_{Pa}$ protein-specific polyclonal antibody from rabbit (Cedarlane) at a 1:1000 dilution in TBST. Blots were probed with either a goat α-mouse or goat α-rabbit HRP-conjugated secondary antibody (Bio-Rad) at 1:300 dilution in TBST for 1 h at 21 °C. Blots were washed five times in TBST. Protein bands were detected using the Super Signal West Pico chemiluminescent substrate from Pierce (Thermo Scientific). Blots were imaged using the Chemidoc XRS System (Bio-Rad).

## Antibody production

AlgL was purified as previously described in ref. 25 and antibodies were produced as previously described[16]. AlgX was purified as previously described in ref. 18 and antibodies were produced as previously described in ref. 16. A construct of AlgK from *P. aeruginosa* (Table S2) was purified as described in this study and antibodies were produced as previously described in ref. 16. AlgG was purified as described previously in ref. 20 and antibodies were produced as previously described in ref. 16. Monoclonal antibody against bacterial RNA polymerase beta (rpoB) was purchased from Invitrogen (RRID: AB_795355).

## Sequence alignment

Protein sequences were obtained from UniProt (https://www.uniprot.org/). UniProt entry numbers for *P. aeruginosa* AlgK, *P. putida* AlgK, *P. aeruginosa* AlgX, and *P. putida* AlgX are P96956, Q88NC7, Q51372, Q88ND0. Pairwise alignments were conducted using EMBOSS Needle[72].

## Reporting summary

Further information on research design is available in the Nature Portfolio Reporting Summary linked to this article.

## Data availability

All data described are located within the manuscript and the supplemental information. Source data are provided with this paper. The coordinates and structure factors for the $AlgKX_{Pp}$ complex have been deposited in the PDB, code 7ULA. Source data are provided with this paper.

## Code availability

The code used to generate the ESI-MS data can be found on GitHub (https://github.com/pkitov/CUPRA-SWARM).

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

## Acknowledgements

Crystallization utilized by the Structural and Biophysical Core Facility at The Hospital for Sick Children, supported in part by the Canadian Foundation for Innovation. Crystallographic data collection utilized the CMCF-BM (08B1-1) beamline at the Canadian Light Source. We thank Dmitri Segal, Jaime Van Loon, and Erum Razvi for their helpful discussions.

This work was supported in part by grants from the Canadian Institutes of Health Research (CIHR) to P.L.H. (MOP 43998 and FDN154327). P.L.H. was the recipient of a Tier 1 Canada Research Chair (2006–2020). This research has been supported by graduate scholarships from Cystic Fibrosis Canada (A.A.G. and G.B.W.); The Hospital for Sick Children Foundation Student Scholarship Program (A.A.G.); the GlycoNet Summer Awards Program for Undergraduate Students (Y.E.L.); and the Natural Sciences and Engineering Research Council of Canada (NSERC) (G.B.W. and J.S.K.); D.T.B. was supported by the Canada Excellence Research Chairs Program (L. Mahal CERC in Glycomics); J.D.C.C. was supported by the Netherlands Organization for Scientific Research (NWO-Vidi).

## Author contributions

A.A.G. and P.L.H. conceptualization, writing, and formal analysis; P.L.H. funding acquisition and project administration; A.A.G., Y.E.L., D.T.B., E.N.K., R.P., K.E.L., M.T.C.W., Q.Z., and G.B.W investigation; J.S.K., J.D.C.C., and P.L.H. resources and supervision.

## Competing interests

The authors declare no competing interests.
