## [Peer Review File · Nature Communications]

Structure of the AlgKX modification and secretion complex required for alginate production and biofilm attachment in *Pseudomonas aeruginosa*Reviewer #1 (Remarks to the Author):

The authors present the structure of the *Pseudomonas putida* periplasmic AlgX-AlgK complex of the two members of alginate biosynthesis and secretion apparatus. The structure of the complex suggests a continuous alginate binding path along both proteins. Further modeling of the AlgXK with outer membrane porin AlgE suggests how the alginate polymer is directed into AlgE. The N-terminus of AlgX is essential for the association with AlgK and disruption of the AlgX-AlgK complex in vivo results in a significant reduction of alginate secretion. Furthermore, the authors provide some data suggesting that AlgXK complex binds preferentially mixed MG polymer over M- or G-homopolymer.

This structural and functional data presented in the manuscript provide a new insight into the last steps leading to secretion of the synthesized and modified alginate from the periplasm to the cell surface.

There are, however, questions that should be additionally addressed in this manuscript. The authors indicate that the contacts between AlgX and AlgK are to a large extent through mainchain-mainchain interactions, with a major contribution from an N-terminus of AlgX, which is disordered in AlgX on its own. What, therefore, contributes to this interaction specificity? Is there a substantial shape complementarity of the two interfaces? The residues involved in the AlgX-AlgK contacts are not well conserved but maybe there are compensatory changes in both proteins in other bacterial strains, especially for the sidechains involved in the interactions? This could be investigated with AlphaFold-created models of AlgXK pairs from different related organisms. I would imagine that the ordered parts of the two proteins form the initial contact point that is next strengthened by wrapping of the AlgX N-terminus around AlgK.

The presented data show that the formation of the complex is necessary for an efficient secretion of alginate. Is the activity of AlgX also essential for secretion? Would an inactive AlgX mutant support secretion to a similar extent as a wild type AlgX?

The data in Table 1 are interpreted as indicative of a preferential binding of GM copolymer to AlgXK complex. I am not convinced by the data. First, the M-oligosaccharides used in these experiments are longer than GM-oligosaccharides, which makes a direct comparison of their K_a values not straightforward. Second, the values of K_a are roughly the same, and the differences for M10 (5.9×10^2) vs M11 (9.3×10^2) vs M12 (4.3×10^2) indicate difficulty in their interpretation. The authors show that AlgK does not coIP with AlgG and they state that the two proteins do not associate. However, AlgG might require a preassembled AlgXK complex or just AlgX only for binding. Is AlgX expressed in these bacterial mutants?

Minor comments:

Figure 1d: no red dashed line for salt bridge, it is marked yellow.

Line 112-113: R3-R10 constitutes eight TPR motifs not nine.

Line 969: D.T.B not D.B.T.

Reviewer #2 (Remarks to the Author):

The work by Gheroghita et al. describes the structure of the AlgXK complex from *Pseudomonas putida* that reveals important insight into the modification and export apparatus of synthase-dependent secretion systems. This work required the crystallization of these proteins from a different species (previously only *P. aeruginosa* and *fluorescens* had been used) and is a remarkable feat in itself, as it delineates for the first time important residues in the binding surface (which previously had been a mystery) and uncovers an electronegative groove that further supports the biological relevance of this interaction in the export of alginate. Importantly, the manuscript also outlines the functional relevance of this interaction by determining the association constants of the proteins for each other and for alginate-specific (M and MG) polymers. In a feat of biological strength, these researchers then systematically underscored these results using enzymatic (acetyltransferase), protein-protein interaction (co-immunoprecipitation) and phenotypic mutant (involving complementation with protein variants) assays. Together, these results substantially advance our understanding of protein complex formation (AlgXK), the binding/secretion of alginate (polymer type, the importance of the N-terminus of AlgX and binding surface extension across AlgXK) and the order of modification (epimerization prior to acetylation)

prior to export.

Revision suggestions:

Main Comments:

1. For the AlgX N-terminal interactions (Lines123-128), the AlgX side chains of E35 and C37 do not appear to be conserved based on your coloring scheme. For AlgK, the conservation (both main and side chain) seems more clear (aside from 404). Since these side chains are not conserved, what does this mean for the relevance of the side-chain interactions in this model? Despite not being conserved are there similar possible residues that could form a link at this position in other *Pseudomonas* species sequences? As it stands, the lack of conservation calls into question the relevance of these side chains and may need some rewording to account for this.....but I should mention that the other conserved residues outweigh the discrepancy with these few residues. On L127, the use of the term "variably conserved" might also need changing since it could be a bit misleading given the side chain conservation issue. L127-128 – "the side chain of S234 and main chain of Y235 interact with A" - Again, the relevance of this needs to be underscored by the conservation coloring scheme that you have used. It seems that Y235 main chain is the most important of these given the conservation?

2. Poly MG Association Constants (L183-184) - I agree that these numbers are roughly double that of Poly M for 2 of the 3 (ie. not GMGGM). This is above the standard error and appears to be significant. However, you might want to be careful in over-interpreting this difference of in vitro numbers that are less than an order of magnitude different. It should also be kept in mind that the polymer lengths are different and it would have been more ideal to test the same lengths. I understand that these polymers can be problematic to make/isolate and it might not always be possible to do this experiment ideally, but perhaps some wording to account for this variability would be beneficial for future readers. As noted later in the document (Discussion), the results that the acetyltransferase activity was not affected by polyM or polyMG substrates supports the need to not over-interpret the association constant results. Some minor rewording in key places could easily emphasize this and shouldn't be a major issue. Once things show up in the literature it can be hard to reconcile with further contradictory data, so being mindful of this with the wording should help to avoid issues in the future.

Other Specific Minor Comments:

L51– biofilm attachment – this might be personal preference, but biofilm formation seems more appropriate

L79-81 - a mention here about the known/unknown 1) levels of acetylation and epimerization; and 2) order to the modifications would be beneficial to keep the current context for the reader. This also will help to place the depiction of Fig. 1a in the proper context.

L87-88 - I agree that there is a gap. This introduction is fairly brief and to the point (which I like), but I think that it could be expanded here with some details as to what is known about AlgK and AlgX in the complex a bit more. A couple lines (or a short paragraph) summarizing the relevant literature on these proteins would be sufficient to lay the foundation for past work that has left the gap that this manuscript will fill.

L99-100 – suggest removing the closing sentence, as it is a bit redundant by the end of the paragraph

L112-113 – model 9 TPRs (R3-10) – this seems like 8 TPRs and looks like it in Figure 1c

L119 – "its" should be "it's" and there is a missing "that" after the (residues 30-37)

L120 – How was the interface calculated? I assume it was with PISA based on the methods, but could you add the program in here to for clarity for the reader.

L184 – remove GMGGM from the text as the two association constant values are here are for the other MG polymers and not for this one.

L195 - This is a great graph, but out of interest sake....when the rates of K and X alone are added

together, does that equal KX? What about for the separate trails of polymer+K+X, does that equal KX in the presence of polymer. Is this really an increase or just an additive amount?

L207 – suggest replacing “Thus” with “Nonetheless” for better flow

L229 – This is more because I am familiar with these plasmids and was interested in the expression system, but how is AlgX making it to the periplasm if the N-terminus is missing? It looks like the expression background (as per the methods section L659) is pET24b and not pET26b (which would have the pelB signal sequence). Perhaps a bit more clarity here would help.

L285 – “our model suggests that polymer acetylation immediately precedes export” - At least for AlgX, but it seems that there is still uncertainty as to how AlgIJF fit into the picture. Could this be earlier or at a different site in the export process? The way this statement reads is that AlgIJF are not part of the picture.

L285-288 – This is only a comment, but this sentence sums up very strong evidence in support of the model and provides great insight into this process. Definitely a worthwhile finding.

L300 – “preferentially binds polyMG ligands” - As noted above.....Some, but not GG ligands?

L319 – I am still not sure that this is a large difference, although polymer binding of itself is significant, the fact that the rate of acetylation wasn't affected suggests that both polyM and MG are within the same binding range of each other to be biologically equal. It is the order within the complex that likely determines that AlgG comes before AlgX activity. Especially given the new information that AlgK does not bind AlgG.

L346 – This is only a suggestion, but starting a new paragraph at “At the inner membrane” might be helpful with the flow for the reader.

L624 – reword “flanking the outside, flanking regions”

Figure 1 – “hydrogen bonds and salt bridge interactions represented by yellow and pink lines” – The pink lines are not evident here. Perhaps adjusting the text or the image would help.

Figure 2 - In panel D, Can the sizing of the gels be made to more closely match the panels in B for easier comparison sake. Perhaps this was shifted in the PDF version. Also, this is a suggestion, but the use of color in the figure is not entirely necessary. For the chromatograms, the use of solid and dashed lines would be as effective and may be clearer to colorblind readers.

Figure 3C - Bars should extend under the PolyM alone and PolyMG alone at the bottom of the graph to denote the series better.

Figure 4a - This is a bit unclear since AlgL and AlgX are similar in size and you can't see both bands on the blots above. Perhaps this just needs some clarification.

Extended Data Table 1 - A couple of things:

1. Tables should have a clear title. The “values in parentheses” is better in the legend below the table.

2. Referencing the formulas used for RMerge and RWork calculations would be nice

3. I assume that you have listed the number of unique reflections here, can you verify this and make it clearer? Also, how many were in the high resolution shell?

4. The Ramachandran statistics (favored and allowed) are mentioned in the text, but often they are included here too.

I know this is an extended table but including the standardized parameters that are commonly listed in Table 1 would be beneficial.

Supplementary Fig. 3&4 - This likely only bothers the chemists among us, but the bond lengths between the C1 and C4 linkages are not consistent. For example, in figure 3 between every third and fourth sugar (starting from the left) the linkage is off and leads to a structure that at first glance is a bit skewed. I suspect that this figure was made by creating a small multimer that was then copied and joined. No problem with that as long as the bond lengths/angles are consistent. Can this be fixed? Also, the C2-OH group runs into the ring oxygen and is hard to read.

Reviewer #3 (Remarks to the Author):

The alginate exopolysaccharide plays a crucial role in the infection by *P. aeruginosa*. In this manuscript, Gheorghita et al. reported the crystal structure of the AlgKX complex from

Pseudomonas putida involved in alginate exopolysaccharide acetylation and export; they found The N-terminus of AlgX is required for complex formation with AlgK, and that formation of the complex is vital for polymer production and biofilm attachment; they further demonstrate that AlgKX preferentially binds modified alginate oligosaccharides. Overall, the authors did a solid job in structure determination of the AlgKX complex and functional analysis of the AlgK-AlgX interactions in alginate production and substrate preference. However, the reviewer thinks the following points need to be addressed by the authors.

1. The authors reported the structure of AlgKX from *P. putida*, yet functional analysis was conducted in *P. aeruginosa*, sequence alignments of the two proteins from different species should be included, at least, in supplementary data.
2. Line 196, how to explain that AlgKX prefers to bind polyMG yet not increase the acetyltransferase activity? Does polyMG promote the formation of AlgKX complex?
3. To obtain the crystal of the AlgKX complex, the authors mixed the purified AlgK and AlgX at a 1:1 molar ratio before crystallization. Just curious, no further size exclusion chromatography is needed? Co-expression and co-purification of the AlgKX complex are not successful?
4. The AlgKX complex is mainly formed by the interaction between the N-terminus of AlgX (residues 30-37) and the TPRs R9-R10 of AlgK. To investigate the important role of the N-terminus residues of AlgX, some point mutations of this N-terminus could be tested to pinpoint the most critical residues in this region for the interaction.
5. Fig. 2a, there are two peaks in the gel filtration chromatograms of AlgKX, and the SDS gel shows that AlgX and AlgK are both in the two peaks. It makes more sense if peak 2 is the excessively one sample of either AlgX or AlgK.
6. Fig. 2d, the migration of AlgX Δ Nterm and AlgK on SDS-gel are same. It is better to include Western blots to show AlgX Δ Nterm and AlgK, respectively.
7. Line 250, to conclude that the formation of the AlgKX complex is required for alginate production, the authors need to exclude the possibility that the AlgX Δ Nterm losing its acetyltransferase activity affects the alginate production.
8. Fig. 3b, the peaks and labels of AlgKX-ligand in ESI mass spectra are confused. The red label is "AlgKX", but the black numbers may indicate the peaks of AlgKX and the red circles indicate the peaks of AlgKX-ligand.
9. The authors used AlphaFold2 to generate a model of the AlgEK. Does AlphaFold2 predict the AlgKX structure with high confidence?
10. In the proposed model of the AlgEKX modification and secretion complex, where is the energy for alginate secretion from? The authors should speculate this, at least in discussion.
11. Extended Data Table 1, numbers in parentheses are not stated.

Response to reviewers' comments:

Reviewer #1:

The authors indicate that the contacts between AlgX and AlgK are to a large extent through mainchain-mainchain interactions, with a major contribution from an N-terminus of AlgX, which is disordered in AlgX on its own. What, therefore, contributes to this interaction specificity? Is there a substantial shape complementarity of the two interfaces? The residues involved in the AlgX-AlgK contacts are not well conserved but maybe there are compensatory changes in both proteins in other bacterial strains, especially for the sidechains involved in the interactions? This could be investigated with AlphaFold-created models of AlgXK pairs from different related organisms. I would imagine that the ordered parts of the two proteins form the initial contact point that is next strengthened by wrapping of the AlgX N-terminus around AlgK.

Response:

*As suggested, we have generated AlphaFold2 models of AlgXK pairs from *P. aeruginosa* and *P. syringae* to investigate their predicted interface interactions. Briefly, the models have the same predicted interaction interface and strongly resemble our *P. putida* AlgXK crystal structure, with Calpha RMSDs < 1.00 Å.*

*In the *P. aeruginosa* and *P. syringae* AlphaFold2 models, the N-terminus of AlgX only uses main chain atoms for the interaction with AlgK. However, we noticed that for all three *Pseudomonas* species, AlgX residues that are outside of the N-terminus mainly interact with AlgK using their side chain atoms. Although the interface is predicted to be the same across all three species, the residues do differ, especially outside of the N-terminus of AlgX. For example, in *P. putida* AlgX, K213 is involved in using its side chain to interact with AlgK. However, this residue is not conserved in *P. syringae* AlgX (I211). Instead, there appears to be a compensatory interaction in *P. syringae* that occurs slightly downstream with the side chain atoms of R214. R214 interacts with AlgK Y318, and this interaction is unique in *P. syringae* as it is not seen in *P. putida* or *P. aeruginosa*. Outside of the N-terminus of *P. aeruginosa* AlgX, there are additional residues, E166 and Y237, that use their side chain atoms to interact with the side chain atoms AlgK R331 and R357. These interactions are unique in *P. aeruginosa* as they are not observed in *P. putida* or *P. syringae*.*

*Given that the N-terminus of AlgX uses almost exclusively main chain atoms for the interaction while residues outside of the N-terminus use side chain atoms for the interaction across all three species, we speculated that the side chains of residues outside the N-terminus of AlgX form the initial contact point with AlgK and thus confer the specificity of the interaction. However, in the *P. syringae* AlgXK model, there are no side chain-side chain interactions that mediate this interaction, which is most likely insufficient to confer interaction specificity. Thus, we assessed the surface hydrophobicity of the interaction interface for all three complexes, revealing a highly conserved and hydrophobic patch of the AlgX N-terminus that is buried in a conserved, deep hydrophobic groove in AlgK. The hydrophobic groove in AlgK is predominantly composed of isoleucine, leucine, and valine; residues that have been previously established to form hydrophobic clusters and mediate protein-protein interactions. This is observed across all three complexes, suggesting that this conserved hydrophobic interaction is responsible for conferring interaction specificity and that interactions outside the N-terminus add additional context to these interactions in certain species.*

We have generated an additional supplementary figure (Supplementary Fig. 5 and 6) to demonstrate this and have expanded on this idea in the results section. Please see lines 135-171 in the main text.

Reviewer #1:

The presented data show that the formation of the complex is necessary for an efficient secretion of alginate. Is the activity of AlgX also essential for secretion? Would an inactive AlgX mutant support secretion to a similar extent as a wild type AlgX?

Response:

*We have shown previously that the activity of AlgX is not essential for secretion of alginate (PMID: 23779107). In our previous study, we determined structure of AlgX, identified its active site residues and mechanism of action. Mutation of these residues rendered the enzyme inactive in vitro. When the same AlgX mutations were introduced onto the chromosome of *P. aeruginosa*, the bacteria still produced alginate comparable to wild type but the acetylation of the polymer was abrogated, as AlgX activity is required for alginate acetylation both in vitro and in vivo. We have included this in the main text at lines 291-295.*

Reviewer #1:

The data in Table 1 are interpreted as indicative of a preferential binding of GM copolymer to AlgXK complex. I am not convinced by the data. First, the M-oligosaccharides used in these experiments are longer than GM-oligosaccharides, which makes a direct comparison of their K_a values not straightforward. Second, the values of K_a are roughly the same, and the differences for M10 (5.9×10^2) vs M11 (9.3×10^2) vs M12 (4.3×10^2) indicate difficulty in their interpretation.

Response:

To respond to the query, we performed additional MS experiments with polyM oligosaccharides that are of comparable length to the polyMG oligosaccharides for a direct comparison of their binding constants (Table 1). After careful examination of all the ESI-MS data, we agree with the reviewer that, when comparing binding across different ligands, the K_a 's are quite similar. Thus, it is difficult to compare binding across different ligands. We have modified our interpretation of the data to simply state that binding was detected for the proteins with the ligands tested. Please see lines 232-247.

Reviewer #1:

The authors show that AlgK does not coIP with AlgG and they state that the two proteins do not associate. However, AlgG might require a preassembled AlgXK complex or just AlgX only for binding. Is AlgX expressed in these bacterial mutants?

Response:

Yes, in the PAO1 Δ_{wspF} P_{BADalg} Δ_{algG} strain with VSV-G-tagged $AlgG_{Pa}$ and PAO1 Δ_{wspF} P_{BADalg} Δ_{algK} strain with VSV-G-tagged $AlgK_{Pa}$, AlgX expression is unaffected by the mutations and is expressed. For PAO1 Δ_{wspF} P_{BADalg} Δ_{algK} strain with VSV-G-tagged $AlgK_{Pa}$, this is shown in (now) Figure 4. We have generated an additional Western blot (Figure 4c) to demonstrate that AlgX is expressed in PAO1 Δ_{wspF} P_{BADalg} Δ_{algG} strain with VSV-G-tagged $AlgG_{Pa}$. The results section has been updated at lines 305-306.

Reviewer #1:

Figure 1d: no red dashed line for salt bridge, it is marked yellow.

Response:

The correction has been made.

Reviewer #1:

Line 112-113: R3-R10 constitutes eight TPR motifs not nine.

Response:

The correction has been made.

Reviewer #1:

Line 969: D.T.B not D.B.T

Response:

The correction has been made.

Reviewer #2:

1. For the AlgX N-terminal interactions (Lines123-128), the AlgX side chains of E35 and C37 do not appear to be conserved based on your coloring scheme. For AlgK, the conservation (both main and side chain) seems more clear (aside from 404). Since these side chains are not conserved, what does this mean for the relevance of the side-chain interactions in this model? Despite not being conserved are there similar possible residues that could form a link at this position in other Pseudomonas species sequences? As it stands, the lack of conservation calls into question the relevance of these side chains and may need some rewording to account for this.....but I should mention that the other conserved residues outweigh the discrepancy with these few residues.

Response:

Reviewer #1 has raised a similar concern. Please see above our response to Reviewer #1 on this issue.

Reviewer #2:

On L127, the use of the term “variably conserved” might also need changing since it could be a bit misleading given the side chain conservation issue.

Response:

We have removed this sentence and dedicated a separate paragraph to discuss the conservation. Please see lines 135-171.

Reviewer #2:

L127-128 – “the side chain of S234 and main chain of Y235 interact with A” - Again, the relevance of this needs to be underscored by the conservation coloring scheme that you have used. It seems that Y235 main chain is the most important of these given the conservation?

Response:

We have attempted to address this in response to Reviewer #1’s comment. Please see above.

Reviewer #2:

2. Poly MG Association Constants (L183-184) - I agree that these numbers are roughly double that of Poly M for 2 of the 3 (ie. not GMGGM). This is above the standard error and appears to be significant. However, you might want to be careful in over-interpreting this difference of in vitro numbers that are less than an order of magnitude different. It should also be kept in mind that the polymer lengths are different and it would have been more ideal to test the same lengths. I understand that these polymers can be problematic to make/isolate and it might not always be possible to do this experiment ideally, but perhaps some wording to account for this variability would be beneficial for future readers. As noted later in the document (Discussion), the results that the acetyltransferase activity was not affected by polyM or polyMG substrates supports the need to not over-interpret the association constant results. Some minor rewording in key places could easily emphasize this and shouldn't be a major issue. Once things show up in the literature it can be hard to reconcile with further contradictory data, so being mindful of this with the wording should help to avoid issues in the future.

Response:

Reviewer #1 raised a similar concern. Please see above our response to Reviewer #1 on this issue.

Briefly, we have performed additional experiments with polyM ligands of comparable lengths to the polyMG ligands. After careful consideration of all the ESI-MS data, we have changed our interpretation of the data. We state whether or not binding was detected, however we are unable to compare binding across different ligands. We further state that we have demonstrated for the first time that a polymer-modifying enzyme-TPR-containing protein complex directly interacts with their relevant exopolysaccharide. Please see lines 232-247.

Reviewer #2:

L51– biofilm attachment – this might be personal preference, but biofilm formation seems more appropriate

Response:

The crystal violet microtitre plate assay used in the manuscript has been previously used to describe biofilm formation however, the biofilm is dumped out of the plate prior to staining. Thus, the assay measures the amount of adherent biofilm biomass that remains in the plate after dumping out the bacterial contents and therefore is a direct measure of the biofilm biomass adherence to the plate. We feel that biofilm adherence/attachment to the plate is a more appropriate description of the readout of this assay.

Reviewer #2:

L79-81 - a mention here about the known/unknown 1) levels of acetylation and epimerization; and 2) order to the modifications would be beneficial to keep the current context for the reader. This also will help to place the depiction of Fig. 1a in the proper context.

Response:

We have implemented the reviewer's suggestion and have included on lines 77-82 the following:

“After translocation, the polymer is modified either by acetylation by the concerted action of AlgI, AlgJ, AlgF, and the terminal acetyltransferase AlgX^{18,19}, or by epimerization to L-gulonate (GulA) by AlgG^{13,20} (Fig. 1a and b). The degree of alginate acetylation can vary from 4 to 57%, depending on the strain of Pseudomonas, its growth conditions, and amount of ManA

within the polymer^{21,22}, while the degree of alginate epimerization is not well-characterized. To date, the order in which these modifications occur is still unknown.”

Reviewer #2:

L87-88 - I agree that there is a gap. This introduction is fairly brief and to the point (which I like), but I think that it could be expanded here with some details as to what is known about AlgK and AlgX in the complex a bit more. A couple lines (or a short paragraph) summarizing the relevant literature on these proteins would be sufficient to lay the foundation for past work that has left the gap that this manuscript will fill.

Response:

We have implemented the reviewer’s suggestion and have added on lines 91-93 the following information summarizing what was currently known about the AlgKX complex prior to our studies presented here.

*“An interaction between AlgX and AlgK has been previously reported in *P. aeruginosa*²⁶, however how these two proteins interact and the consequences of this interaction on alginate production have not been investigated.”*

Reviewer #2:

L99-100 – suggest removing the closing sentence, as it is a bit redundant by the end of the paragraph

Response:

We have removed the sentence as per the reviewer’s suggestion.

Reviewer #2:

L112-113 – model 9 TPRs (R3-10) – this seems like 8 TPRs and looks like it in Figure 1c

Response:

The correction has been made.

Reviewer #2:

L119 – “its” should be “it’s” and there is a missing “that” after the (residues 30-37)

Response:

The correction has been made.

Reviewer #2:

L120 – How was the interface calculated? I assume it was with PISA based on the methods, but could you add the program in here to for clarity for the reader.

Response:

The reviewer is correct – PISA was used to calculate the interface buried surface area. This following information has been added on lines 123-128 to clarify this issue:

“Using the Proteins, Interface, Surfaces, and Assemblies (PISA) server²⁴, the solvation energy and total binding energy of this interface were calculated to be -7.874 kcal/mol and -13.57 kcal/mol, respectively, suggesting that the interface we observe is biologically relevant and not

an artifact of crystal-packing. PISA also calculated the interaction interface to have a buried surface area of 1015 Å² mediated by 12 hydrogen bonds and one salt bridge.”

Reviewer #2:

L184 – remove GMGGM from the text as the two association constant values are here are for the other MG polymers and not for this one.

Response:

We removed the association constants. We also noticed that we had an error in the manuscript – the correct ligand name should be GMGGMG, not GMGGM. This has been corrected wherever necessary throughout the manuscript.

Reviewer #2:

L195 - This is a great graph, but out of interest sake....when the rates of K and X alone are added together, does that equal KX? What about for the separate trails of polymer+K+X, does that equal KX in the presence of polymer. Is this really an increase or just an additive amount?

Response:

On review of the data, we concur with the reviewer that the AlgKX increase appears to be additive. Thus, we do not believe that the observed increase in acetylcysteine activity is a true increase due to complex formation. Therefore, we do not find it informative to compare activity of AlgX alone to AlgXK. Additionally, the high background levels of polyM on its own also suggest to us that the observed increase when polyM is added to either AlgK, AlgX, or AlgKX may also be additive. We have adjusted this section of the results to only compare how the activity of X alone is influenced by polyMG, or how the activity of XK is influenced by polyMG, as polyMG has a low background signal. However, for transparency, we present all the data and address these issues. Lines 232-247 in the revised manuscript now read as follows:

“TPR-containing proteins have previously been demonstrated to influence the activity of polymer modifying enzymes in vitro in the Pel biosynthetic system²⁸. Thus, we next assessed whether the presence of AlgK_{Pp} influences the in vitro enzymatic activity of AlgX_{Pp}. Initially, we monitored the rate of acetylcysteine activity – the ability to remove acetyl groups – of AlgK_{Pp}, AlgX_{Pp} and AlgKX_{Pp} using 4-nitrophenyl acetate as an acetyl group donor with removal of the acetate being monitored at 405 nm in real time. Although it appears that the AlgK_{Pp}-AlgX_{Pp} interaction increased acetylcysteine activity compared to AlgX_{Pp} alone, we believe this is an additive effect due to the unexpectedly high background of AlgK_{Pp} and therefore not a true increase in activity as a result of complex formation (Supplementary Fig. 11). We next assessed whether the presence of either polyM or polyMG – the acetyl group acceptor – influenced AlgX_{Pp}'s acetylcysteine activity. Considering the high background of polyM on its own, we speculate that the observed increase upon addition of polyM to either AlgK_{Pp}, AlgX_{Pp}, or AlgK-AlgX_{Pp} is also due to an additive effect. Thus, we are unable to draw any conclusions regarding the effect of polyM on enzyme activity. However, addition of polyMG to either AlgX_{Pp} or AlgKX_{Pp} results in significantly increased acetylcysteine activity, demonstrating that presence of an acetyl group acceptor influences AlgX_{Pp} activity (Supplementary Fig. 11).”

Reviewer #2:

L207 – suggest replacing “Thus” with “Nonetheless” for better flow

Response:

We have implemented the reviewer's suggestion.

Reviewer #2:

L229 – This is more because I am familiar with these plasmids and was interested in the expression system, but how is AlgX making it to the periplasm if the N-terminus is missing? It looks like the expression background (as per the methods section L659) is pET24b and not pET26b (which would have the pelB signal sequence). Perhaps a bit more clarity here would help.

Response:

L229 refers to P. aeruginosa AlgX^{ΔNterm} being expressed at the attTn7 site on the P. aeruginosa chromosome. In this construct of P. aeruginosa AlgX, the N terminal deletion includes residues 29-41. Thus, AlgX^{ΔNterm}'s native signal sequence (residues 1-26) is still intact and able to traffic the protein to the periplasm, as shown by the Western blot in Fig. 4a. The residues involved in the deletion are detailed in the supplementary Table S1. This information has been added to the results section of the manuscript for clarity at lines 276-277.

In the P. putida AlgX^{ΔNterm} construct, the N terminal deletion (residues 1-38) includes the native signal sequence. The residues involved in the deletion are detailed in the supplementary Table S1. This construct was expressed using pET26b, not pET24b as previously stated, which has a PelB signal sequence. We thank the reviewer for catching this error. This construct of P. putida AlgX^{ΔNterm} was then purified and used for in vitro experiments.

Reviewer #2:

L285 – “our model suggests that polymer acetylation immediately precedes export” - At least for AlgX, but it seems that there is still uncertainty as to how AlgIJF fit into the picture. Could this be earlier or at a different site in the export process? The way this statement reads is that AlgIJF are not part of the picture.

Response:

AlgIJFX form the acetylation machinery in which AlgX is the terminal acetyltransferase. Although AlgJ is also a periplasmic acetyltransferase, we have previously shown that only AlgX can directly bind to alginate and acetylate it (PMID: 25165982). Furthermore, AlgI, AlgJ, and AlgF can be deleted from P. aeruginosa without loss of alginate production, although the alginate is not acetylated. Only deletion of AlgX results in no alginate production. Our current understanding is that AlgI receives an acetyl group from the cytoplasm that it then transfers to AlgJ in the periplasm. AlgJ transfers the acetyl group to AlgX and then finally AlgX transfers the acetyl group to the polymer. The role of AlgF is not understood, but its structure and sequence do not suggest it is an acetyltransferase. AlgX could associate with the rest of the acetylation complex while also interacting the AlgK and the polymer, but how this might occur is not known. We have clarified in the text that we are referring to acetylation by the terminal acetyltransferase AlgX and epimerization by AlgG (line 341).

Reviewer #2:

L285-288 – This is only a comment, but this sentence sums up very strong evidence in support of the model and provides great insight into this process. Definitely a worthwhile finding.

Response:

We thank the reviewer for their comment.

Reviewer #2:

L300 – “preferentially binds polyMG ligands” - As noted above.....Some, but not GG ligands?

Response:

In the second paragraph of the “The AlgKX complex directly binds alginate” results subsection, we indicate that P. aeruginosa does not make polyMG ligands with consecutive GG residues, thus they are not biologically relevant. Please see lines 215-217.

Reviewer #2:

L319 – I am still not sure that this is a large difference, although polymer binding of itself is significant, the fact that the rate of acetylation wasn't affected suggests that both polyM and MG are within the same binding range of each other to be biologically equal. It is the order within the complex that likely determines that AlgG comes before AlgX activity. Especially given the new information that AlgK does not bind AlgG.

Response:

As outlined above, we have changed our interpretation of the polymer binding data. We have also removed our comment in the discussion section that states how preferential binding to polyMG relates to order of modifications.

Reviewer #2:

L346 – This is only a suggestion, but starting a new paragraph at “At the inner membrane” might be helpful with the flow for the reader.

Response:

We have implemented the reviewer's suggestion.

Reviewer #2:

L624 – reword “flanking the outside, flanking regions”

Response:

We have made the correction. We thank the reviewer for their close attention to details.

Reviewer #2:

Figure 1 – “hydrogen bonds and salt bridge interactions represented by yellow and pink lines” – The pink lines are not evident here. Perhaps adjusting the text or the image would help.

Response:

The correction has been made.

Reviewer #2:

Figure 2 - In panel D, Can the sizing of the gels be made to more closely match the panels in B for easier comparison sake. Perhaps this was shifted in the PDF version. Also, this is a suggestion, but the use of color in the figure is not entirely necessary. For the chromatograms, the use of solid and dashed lines would be as effective and may be clearer to colorblind readers.

Response:

The gels were poured by hand, resulting in differences in size and perhaps also differences in how the gels ran. We acknowledge that the use of colour is not entirely necessary, however we think it helps to visually orient the reader by associating the chromatograms with their relevant gels. We have used colourblind-friendly colours so that colourblind readers may still make distinctions between the chromatograms.

Reviewer #2:

Figure 3C - Bars should extend under the PolyM alone and PolyMG alone at the bottom of the graph to denote the series better.

Response:

We have edited the graph and removed the bars.

Reviewer #2:

Figure 4a - This is a bit unclear since AlgL and AlgX are similar in size and you can't see both bands on the blots above. Perhaps this just needs some clarification.

Response:

All blots were done on the same membrane. We first probed for AlgX, then stripped the membrane, and then cut the membrane in half to probe for AlgL and RNAP, separately. Thus, because of the stripping process, we would not expect to observe bands for both AlgX and AlgL in either of the blots. Unfortunately, we forgot to include this important detail in the methods. We have added a detailed description of the Western blot stripping process in the methods section. Please see lines 815-838.

Reviewer #2:

1. Tables should have a clear title. The "values in parentheses" is better in the legend below the table.

Response:

The correction has been made.

Reviewer #2:

2. Referencing the formulas used for RMerge and RWork calculations would be nice

Response:

Nature's Table 1 template indicates that "Equations defining various R-values are standard and hence are no longer defined in the footnotes." Thus, we have omitted the formulas.

Reviewer #2:

3. I assume that you have listed the number of unique reflections here, can you verify this and make it clearer? Also, how many were in the high resolution shell?

Response:

We have updated Extended Data Table 1 (now Supplementary Table 1) to clarify a total number of 38142 unique reflections, with 3741 in the highest resolution shell. In the refinement, 38104 reflections were used and 3715 were in the highest resolution shell.

Reviewer #2:

4. The Ramachandran statistics (favored and allowed) are mentioned in the text, but often they are included here too. I know this is an extended table but including the standardized parameters that are commonly listed in Table 1 would be beneficial.

Response:

We have added the Ramachandran statistics to Supplementary Table 1.

Reviewer #2:

Supplementary Fig. 3&4 - This likely only bothers the chemists among us, but the bond lengths between the C1 and C4 linkages are not consistent. For example, in figure 3 between every third and fourth sugar (starting from the left) the linkage is off and leads to a structure that at first glance is a bit skewed. I suspect that this figure was made by creating a small multimer that was then copied and joined. No problem with that as long as the bond lengths/angles are consistent. Can this be fixed? Also, the C2-OH group runs into the ring oxygen and is hard to read.

Response:

We have made changes to the chemical structures of the oligosaccharides as requested (now in Supplementary Figures 7 and 8).

Reviewer #3:

1. The authors reported the structure of AlgKX from *P. putida*, yet functional analysis was conducted in *P. aeruginosa*, sequence alignments of the two proteins from different species should be included, at least, in supplementary data.

Response:

*As we were unable to obtain crystals with the *P. aeruginosa* AlgKX complex, we felt that it would be best to remain consistent in our *in vitro* experiments and continue to work with the *P. putida* constructs that successfully crystallized and produced a structure. We have included a sequence alignment of the two proteins in the supplementary information as requested – Supplementary Figures 1 and 2.*

Reviewer #3:

2. Line 196, how to explain that AlgKX prefers to bind polyMG yet not increase the acetylcetase activity? Does polyMG promote the formation of AlgKX complex?

Response:

As formation of the AlgKX complex occurs even in the absence of alginate polymer, we do not think that polyMG promotes complex formation. Reviewer #2 also commented on our acetylcetase activity. We have addressed concerns with our assay (please see above comments). Briefly, we are unable to assess addition of polyM binding to the complex due to the high background signal of the polymer. Thus, we cannot compare differences in activity between polyM and polyMG addition to the complex.

Reviewer #3:

3. To obtain the crystal of the AlgKX complex, the authors mixed the purified AlgK and AlgX at a 1:1 molar ratio before crystallization. Just curious, no further size exclusion chromatography is needed? Co-expression and co-purification of the AlgKX complex are not successful?

Response:

As our initial crystallization trials mixing separately purified AlgK and AlgX proteins was successful we, did not attempt co-expression and co-purification of the AlgKX complex and no further size exclusion chromatography of the complex was performed prior to crystallization. As our results demonstrate, this method was sufficient for structural determination of the complex.

Reviewer #3:

4. The AlgKX complex is mainly formed by the interaction between the N-terminus of AlgX (residues 30-37) and the TPRs R9-R10 of AlgK. To investigate the important role of the N-terminus residues of AlgX, some point mutations of this N-terminus could be tested to pinpoint the most critical residues in this region for the interaction.

Response:

The reviewer makes an excellent suggestion. We agree that point mutations of residues in the N-terminus could be tested to determine interaction hot-spot amino acid, but we do not believe that it is required for the story we present in our manuscript.

Reviewer #3:

5. Fig. 2a, there are two peaks in the gel filtration chromatograms of AlgKX, and the SDS gel shows that AlgX and AlgK are both in the two peaks. It makes more sense if peak 2 is the excessively one sample of either AlgX or AlgK.

Response:

We interpret the chromatogram as having one peak and a shoulder to the right of the peak rather than two peaks. AlgX and AlgK are present in both in the peak and the shoulder. The peak represents the AlgKX complex comigrating off the column, and the shoulder represents unbound AlgX and AlgK.

Reviewer #3:

6. Fig. 2d, the migration of AlgX Δ Nterm and AlgK on SDS-gel are same. It is better to include Western blots to show AlgX Δ Nterm and AlgK, respectively.

Response:

Although we agree with the reviewer that a Western blot is ideal, we believe the Coomassie staining is sufficient to demonstrate that AlgX ^{Δ Nterm}-AlgK does not result in comigration off the column, as is observed in Fig. 2a and 2b.

Reviewer #3:

7. Line 250, to conclude that the formation of the AlgKX complex is required for alginate production, the authors need to exclude the possibility that the AlgX Δ Nterm losing its acetylerase activity affects the alginate production.

Response:

We have added a supplementary figure (Supplementary Fig. 14) that demonstrates that AlgX and AlgX Δ Nterm have comparable acetylerase activities in vitro. We have revised the manuscript on lines 303-306 and have stated this result.

Reviewer #3:

8. Fig. 3b, the peaks and labels of AlgKX-ligand in ESI mass spectra are confused. The red label is “AlgKX”, but the black numbers may indicate the peaks of AlgKX and the red circles indicate the peaks of AlgKX-ligand.

Response:

The correction has been made.

Reviewer #3:

9. The authors used AlphaFold2 to generate a model of the AlgEK. Does Alphafold2 predict the AlgKX structure with high confidence?

Response:

Yes. We have added a supplementary figure (Supplementary Figure 5) that shows the AlphaFold2 models of Pseudomonas syringae and Pseudomonas aeruginosa AlgKX. These models have a Calpha RMSD of < 1.00 Å when superimposed with our Pseudomonas putida AlgKX structure.

Reviewer #3:

10. In the proposed model of the AlgEKX modification and secretion complex, where is the energy for alginate secretion from? The authors should speculate this, at least in discussion.

Response:

The reviewer asks an excellent question. The energy required for alginate secretion has been previously investigated using steered molecular dynamics simulations with alginate and the structure of AlgE (PMID: 25084326). The authors speculate that alginate does not diffuse passively through the pore of AlgE and that energy is required for alginate export. This is presumed to be provided by the alginate synthesis machinery that extends the polymer. To mimic this, the authors applied a force to either “push” or “pull” alginate (a polyMG polymer) through the pore of AlgE from the periplasmic side to the extracellular side and vice versa. They concluded that AlgE itself does not appear to impart any directionality to alginate transport. When a simulation was done with alginate starting in pore of AlgE, alginate twisted and turned through the pore, possible aided by “breathing” motions of the protein. This was not observed when a simulation was done in with AlgE conformation was fixed.

Similar simulations could be done in the future using our AlgEKX model to see if presence of a modification and secretion complex results in different outcomes.

We implemented the reviewer’s suggestion and touch on this subject in the discussion at lines 414-425.

Reviewer #3:

11. Extended Data Table 1, numbers in parentheses are not stated.

Response:

We have made the correction.

Reviewer #1 (Remarks to the Author):

The authors addressed adequately all my comments and responded to my questions. The revised manuscript is improved over the original submission. I have no further comments.

Reviewer #2 (Remarks to the Author):

I appreciate the time and effort that has gone into addressing all of the reviewer's questions from the first submission. I feel that the manuscript is greatly improved and have only a few further comments listed below:

L133 – In the description of the interactions outside the N-terminus of AlgX the residues R174 and K213 were not mentioned at this point, yet they are depicted in both Figure 1d&e. Adding in a sentence referring to these residues would help to clarify the full summary interactions at this point in the manuscript before later defining the commonalities/differences to the Pa and Ps complexes (2 paragraphs later).

L170 – change AlgKK to AlgKX

L226-228 (L377)– Recognizing that it is difficult to compare different polymers, the significance could also be said of the similarly sized M6 and M7 polymers, although to a lesser degree admittedly. It is interesting to speculate how this would play into AlgG mutants where the polymer is still secreted as polyM...

L245-247 – This data is important and I recognize the difficulty in comparing all of these values, However, is the KX versus KX-polyMG still significant if the average polyMG value is subtracted from the KX-polyMG first? In effect this would “zero” for polyMG activity. It is also a bit confusing that the AlgK versus K-polyMG is significant. Perhaps a similar “zeroing” of this fraction should be done?

L267 – change “interaction requires presence” to “interaction requires the presence”

L807 – Gen is not in the list of abbreviations, but should likely just be spelled out in full

L828 – change room temperature to 22 degrees

Figure 1d&e – This is an aesthetic comment, but the hydrogen bond dashed lines (yellow) are not consistent in these panels. Some appear as dashes and others as solid lines. Please fix the width and radius of the dashed lines so that they appear more consistent.

Reviewer #3 (Remarks to the Author):

In this revised version, the authors addressed my major concerns. I recommend to accept this manuscript.

Reviewer #1: The authors addressed adequately all my comments and responded to my questions. The revised manuscript is improved over the original submission. I have no further comments.

Response: We thank the reviewer for their comment.

Reviewer #2: I appreciate the time and effort that has gone into addressing all of the reviewer's questions from the first submission. I feel that the manuscript is greatly improved and have only a few further comments listed below:

Response: We thank the reviewer for their comment.

Reviewer #2: L133 – In the description of the interactions outside the N-terminus of AlgX the residues R174 and K213 were not mentioned at this point, yet they are depicted in both Figure 1d&e. Adding in a sentence referring to these residues would help to clarify the full summary interactions at this point in the manuscript before later defining the commonalities/differences to the Pa and Ps complexes (2 paragraphs later).

Response: We have added two sentences to describe these interactions at L132-134:

“The side chain of AlgX_{Pp} R174 interacts with the side chain of AlgK_{Pp} N368 and the main chain of G367 via hydrogen bonding. The side chain of AlgX_{Pp} K213 interacts by hydrogen bonding with the main chain of AlgK_{Pp} A333 (Fig. 1d).”

Reviewer #2: L170 – change AlgKK to AlgKX

Response: The correction has been made.

Reviewer #2: L226-228 (L377)– Recognizing that it is difficult to compare different polymers, the significance could also be said of the similarly sized M6 and M7 polymers, although to a lesser degree admittedly. It is interesting to speculate how this would play into AlgG mutants where the polymer is still secreted as polyM....

Response: We agree with the reviewer that the complex does appear to bind several of the ligands more tightly which is why we had phrased the sentence “for some of the ligands tested, including the biologically relevant GMGMGM and GMGMGMG ligands...”. By using the word “including” we are not excluding the binding observed for M6 and M7, we just chose to highlight the biologically relevant GM compounds. We agree with the reviewer, future work investigating if/how alginate secretion via the AlgKX complex is affected in the absence of Gula residues would be interesting.

Reviewer #2: L245-247 – This data is important and I recognize the difficulty in comparing all of these values, However, is the KX versus KX-polyMG still significant if the average polyMG value is subtracted from the KX-polyMG first? In effect this would “zero” for polyMG activity. It is also a bit confusing that the AlgK versus K-polyMG is significant. Perhaps a similar “zeroing” of this fraction should be done?

Response: We have implemented the reviewer's suggestion and added two panels to Supplementary Figure 12 to show the baseline-corrected values. For transparency, panel a shows all the data prior to baseline correction. Panels b and c show the baseline-corrected values where the polyM and polyMG data are defined as the baseline, respectively. Starting at L243, we state the following:

“We baseline-corrected values against the polyM (Supplementary Fig. 12b) and polyMG (Supplementary Fig. 12c) data. Addition of polyM to either AlgK_{pp} or AlgX_{pp} did not significantly increase activity, while addition to AlgKX_{pp} resulted in a significant increase in acetylcysteine activity (Supplementary Fig. 12b). Furthermore, addition of polyMG to AlgK_{pp}, AlgX_{pp}, and AlgKX_{pp} significantly increased activity (Supplementary Fig. 12c). As AlgK_{pp} is not an acetyltransferase enzyme, the observed increase is most likely due to nonspecific hydrolysis of the pseudosubstrate. Overall, the data demonstrate that addition of an acetyl group acceptor, either polyM or polyMG, influences AlgKX_{pp} acetylcysteine activity.”

Reviewer #2: L267 – change “interaction requires presence” to “interaction requires the presence”

Response: The correction has been made.

Reviewer #2: L807 – Gen is not in the list of abbreviations, but should likely just be spelled out in full

Response: We have added gentamicin (Gen) to the list of abbreviations.

Reviewer #2:

L828 – change room temperature to 22 degrees

Response: We have changed “room temperature” to “21 °C” throughout the main text, where appropriate.

Reviewer #2: Figure 1d&e – This is an aesthetic comment, but the hydrogen bond dashed lines (yellow) are not consistent in these panels. Some appear as dashes and others as solid lines. Please fix the width and radius of the dashed lines so that they appear more consistent.

Response: The changes have been made.

Reviewer #3: In this revised version, the authors addressed my major concerns. I recommend to accept this manuscript.

Response: We thank the reviewer for their comment.